# Conditional privatization of a public siderophore enables *Pseudomonas aeruginosa* to resist cheater invasion

Zhenyu Jin[1], Jiahong Li[1], Lei Ni[1], Rongrong Zhang [1], Aiguo Xia[1] & Fan Jin [1,2,3]

Understanding the mechanisms that promote cooperative behaviors of bacteria in their hosts is of great significance to clinical therapies. Environmental stress is generally believed to increase competition and reduce cooperation in bacteria. Here, we show that bacterial cooperation can in fact be maintained because of environmental stress. We show that *Pseudomonas aeruginosa* regulates the secretion of iron-scavenging siderophores in the presence of different environmental stresses, reserving this public good for private use in protection against reactive oxygen species when under stress. We term this strategy "conditional privatization". Using a combination of experimental evolution and theoretical modeling, we demonstrate that in the presence of environmental stress the conditional privatization strategy is resistant to invasion by non-producing cheaters. These findings show how the regulation of public goods secretion under stress affects the evolutionary stability of cooperation in a pathogenic population, which may assist in the rational development of novel therapies.

[1] Hefei National Laboratory for Physical Sciences at the Microscale, University of Science and Technology of China, 230026 Hefei, China. [2] Department of Polymer Science and Engineering, University of Science and Technology of China, 230026 Hefei, China. [3] CAS Key Laboratory of Soft Matter Chemistry, University of Science and Technology of China, 230026 Hefei, China. These authors contributed equally: Zhenyu Jin, Jiahong Li. Correspondence and requests for materials should be addressed to F.J. (email: fjinustc@ustc.edu.cn)

Natural selection and game theory[1–3] predict that cooperative behaviors are vulnerable to the exploitation of common resources by selfish individuals because selfish individuals (cheaters) consume common resources to gain benefits without contributing fairly. In this context, selfish individuals are fitter, eventually resulting in the collapse of cooperation[4] when the magnitude of selfish exploitation exceeds the capacity of a cooperative, a situation known as the 'tragedy of the commons'[5]. By contrast, cooperation mediated by producing public goods is consistently observed in diverse life forms ranging from microbes to social animals; for example, the yeast *Saccharomyces cerevisiae* secretes enzymes to externally digest sucrose and thus facilitates sucrose metabolism at the group level[6], and the ant *Pristomyrmex punctatus* collectively contribute their efforts to build and maintain the nest for their housing[7]. Because of their experimental accessibility and the extensive and diverse genomic data available, microbes are used as model organisms by researchers to probe social interactions at the molecular level and examine fundamental aspects of the origins of cooperation in well-defined systems[3]. In past decades, numerous theoretical and experimental investigations have elucidated several mechanisms and processes that promote cooperation in microbes, including kin selection/discrimination[8], cooperation when surrounded by cooperators/quorum sensing[9,10], punishment/policing[11,12], pleiotropy[13], phenotype variation/bistability[14], spatial self-organization[15], and adaptation to changing environments[16].

Microbes have evolved elaborate strategies to adapt to nutrient scarcity and environmental stresses, which is generally believed to increase competition and reduce cooperation among them. For example, molecular mechanisms underlying decision-making enable bacteria to change their phenotypes or tune their gene expression to adjust to environmental or nutritional perturbations[17,18] and to counter-attack when they sense nutrient limitation or direct cell damage[19]. We hypothesize that environmental stresses that induce competition may also stimulate the cooperator to compete and thereby to resist cheater invasion, thus in turn promoting the evolutionary stability of those cooperative traits. This hypothesis is supported by the findings of Xavier et al.[20] who reported that swarming[21], a form of cooperative motility mediated by rhamnolipid secretion, can be maintained in *Pseudomonas aeruginosa* because carbon-rich rhamnolipid is secreted only when nitrogen is more limiting than carbon.

To elucidate mechanisms that make cooperators robust to environmental stresses, such as starvation, antimicrobial treatment, or oxidative stresses, we investigated how *P. aeruginosa* regulates the production and secretion of iron-scavenging siderophores in the presence of different environmental stresses. In addition, we evaluated whether these environmental stresses can facilitate the persistence of bacterial cooperation in this cooperative system[22,23]. Using the combination of a single-cell tracking technique, experimental evolution, and theoretical modeling, we identified a strategy we term "conditional privatization" that can increase survival and resist invasion by non-producing cheaters in the presence of environmental stresses.

## Results

### Stress induce the pyoverdine accumulation in the periplasm.
We used a spinning-disc confocal microscope with high spatio-temporal resolution to directly image and localize pyoverdine (PVDI), a fluorescent siderophore[24]. We observed that illumination with a violet laser light (405 nm) led to the accumulation of PVDI in the periplasms of single *P. aeruginosa* cells (Fig. 1a–c and Supplementary Movie 1). The PVDI concentration increased approximately eight-fold within 4 min when illuminance

exceeded $6.00\,mW\,cm^{-2}$. Fluorescent PVDIs were excited when illuminated by the 405 nm laser source. To decouple the imaging of PVDI and the light stimulus of cells, we carefully optimized laser power and exposure time to identify an excitation condition that would not lead to the accumulation of PVDI or affect the growth of cells. Figure 1d and Supplementary Figure 1 indicate that relatively weak illumination ($<3.00\,mW\,cm^{-2}$) does not affect bacterial growth and the excitation condition ($0.10\,mW\,cm^{-2}$) does not result in PVDI accumulation. Therefore, $0.10\,mW\,cm^{-2}$ excitation was applied to image PVDI in this study. The accumulation of PVDI in the bacterial periplasm may arise from two possibilities: (i) a specific response to the stimulus of violet/blue light[25] or (ii) a general response to photon stress that typically leads to reactive oxygen species (ROS) generation in live cells[26]. To assess these two possibilities, next, we used a ROS-specific dye (2′,7′-dichlorofluorescin diacetate, $H_2DCFDA$) in situ to monitor the intracellular levels of ROS in single cells[27] stimulated with violet light ($\geq 6.00\,mW\,cm^{-2}$). This illumination led to the accumulation of PVDI in 2 min. ROS was generated within 8 min in single *P. aeruginosa* cells (Fig. 1e and Supplementary Movie 2), as indicated by bluish green to show green fluorescence, after stimulation with violet light ($\geq 9.00\,mW\,cm^{-2}$). This finding suggests that the accumulation of PVDI is a general response to photon stress rather than a specific consequence of the stimulus of violet/blue light. To further confirm this finding, we used a low dose of tobramycin, an aminoglycoside antibiotic, with concentrations near the minimal inhibitory concentration to treat *P. aeruginosa* to enable ROS generation[28]. A low dose of tobramycin also led to the accumulation of PVDI through ROS generation in cells (Fig. 1f). This result indicates that different environmental stresses, including photon and antimicrobial stresses, trigger the accumulation of PVDI in the periplasms of *P. aeruginosa*.

### Accumulated PVDI enables survival of bacteria under stresses.
It has been reported that siderophore-mediated iron acquisition is required for resistance to ROS stress in the fungus *Alternaria alternate*[29]. This report motivated us to investigate the mechanism underlying the accumulation of PVDI in *P. aeruginosa* cells in response to ROS generation. Intracellular reactive species of superoxide ($O_2\cdot$) or peroxide ($H_2O_2$) can damage Fe–S clusters and result in the release of free ferrous ions from the damaged clusters[30]. The released ferrous ions can subsequently catalyze $H_2O_2$ to form the most harmful hydroxyl radical (OH·) through the Fenton reaction[31]. PVDI accumulated in *P. aeruginosa* cells may chelate the released ferrous and thereby repress OH· formation, consequently protecting cells from oxidative damage. To directly validate this conjecture, we used a combination of propidium iodide (PI)-staining and ROS staining to examine whether a mutant strain ($\Delta pvdA$) deficient in PVDI production[32] can tolerate the photon or antimicrobial stress used to treat the wild-type strain. Under the treatment of violet light ($3.00\,mW\,cm^{-2}$), tobramycin ($1.0\,\mu g\,mL^{-1}$) or gentamicin ($2.0\,\mu g\,mL^{-1}$) in iron-limited conditions, a higher percentage of $\Delta pvdA$ cells (18.5% ($p < 10^{-6}$), 23.1% ($p < 10^{-6}$) or 25.3% ($p < 10^{-6}$)) generated ROS than that of wild-type cells (1.2%, 9.9%, or 0.7%) (Fig. 2b, Supplementary Fig. 2c, Supplementary Movie 3) and a higher percentage of $\Delta pvdA$ cells (15.4% ($p < 10^{-6}$), 15.4% ($p < 10^{-6}$) or 8.2% ($p < 10^{-6}$)) were damaged than that of wild-type cells (0.7%, 0.8% or 1.5%) (Fig. 2b, Supplementary Fig. 2d, Supplementary Movie 4). To further explore whether the protective activity of PVDI is due to the repression of the Fenton reaction, we examined whether PVDI can protect the wild-type and $\Delta pvdA$ strains from antimicrobial stress in iron-rich conditions. Our results indicate that PVDI does not protect cells by preventing ROS

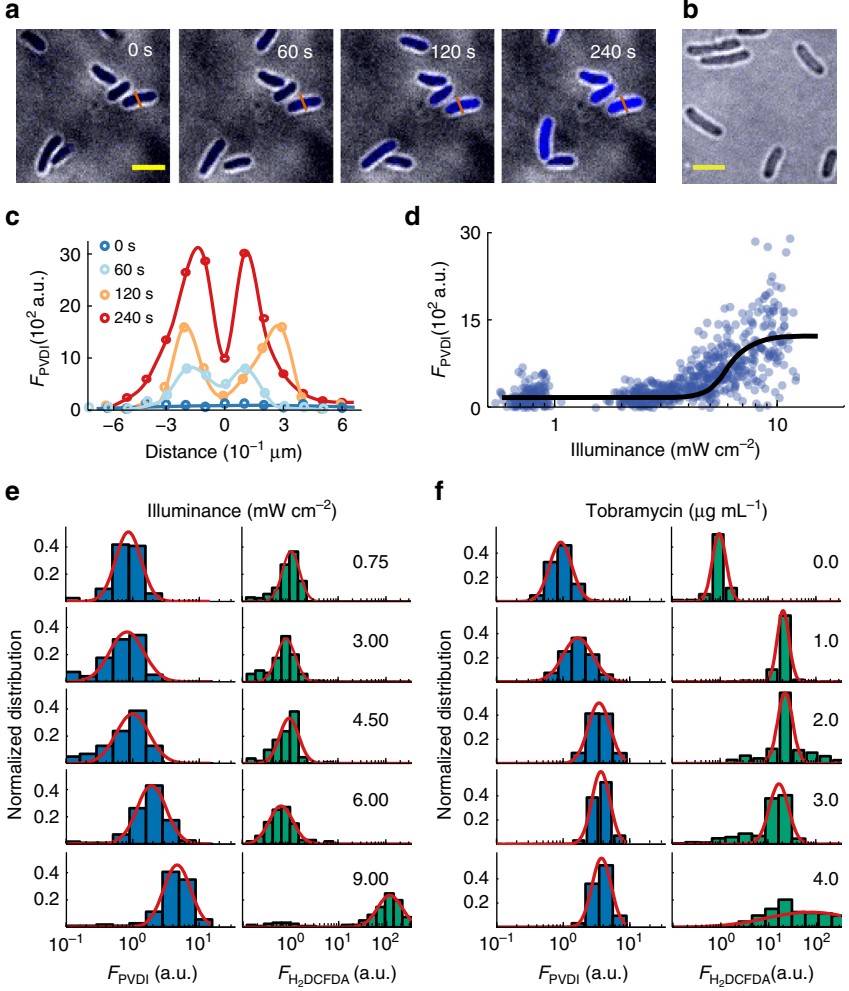

**Fig. 1** Environmental stresses trigger the accumulation of a bacterial siderophores in the periplasms of *P. aeruginosa*. Representative bright-field (gray) + confocal images show that **a** illuminations of violet-laser (405 nm, 6.00 mW cm$^{-2}$) trigger the accumulation of a bacterial siderophores (pyoverdine, PVDI) in wild-type *P. aeruginosa* within 4 min and **b** Δ*pvdA* cells did not produce PVDI despite illuminations (6.00 mW cm$^{-2}$), where the PVDI is indicated by blue colors. **c** Distributions of PVDI fluorescence intensities along the vermillion line across the bacterium shown in **a** for the time $t = 0, 60, 120, 240$ s, from bottom to top. **d** Photon-stress dependence of PVDI fluorescence intensity in single cells, where cells were exposed to violet-laser for 2 min. The dots and the line present data arising from single bacteria and from the average of multiple bacteria, respectively. **e**, **f** Normalized histograms of PVDI (blue bar) fluorescent intensities or ROS-specific dye H$_2$DCFDA fluorescent intensities (bluish green bar) arising single cell in the presence of **e** different photon-stresses (0.75, 3.00, 4.50, 6.00, and 9.00 mW cm$^{-2}$) or in addition of **f** different amount of tobramycin (0.0, 1.0, 2.0, 3.0, and 4.0 μg mL$^{-1}$), where cells were exposed to violet-laser for 2 or 8 min or tobramycin for 1 or 7 h, respectively. Scale bar for all images are 2 μm

generation or cell damage in the presence of antimicrobial stress in iron-rich conditions (Supplementary Fig. 3a–d). This finding suggests that the chelation of excess iron is essential for the protection of cells from damage; therefore, the protective activity of PVDI declined in iron-rich conditions. These findings fully support our conjecture.

Furthermore, we expected that only PVDI accumulated in bacterial periplasms—not secreted PVDI—can protect cells in the presence of environmental stresses. To test this prediction, we exogenously added PVDI (5.0 μM) in the culture media for Δ*pvdA* cells or a double-mutant strain Δ*pvdA*Δ*fpvA* deficient in the uptake of exogenous PVDI (ePVDI)[33], allowing only the Δ*pvdA* cells to absorb exogenous ePVDI. We observed that after the absorption of ePVDI, the fraction of Δ*pvdA* cells that died (0.8% ($p = 0.2557$) or 1.5% ($p = 0.9725$)) was similar to that of the wild-type strain in the presence of photon (Fig. 2b and Supplementary Movie 5) or antimicrobial (Supplementary Fig. 2b) stress, and the fraction of Δ*pvdA* cells that died was significantly lower than that of Δ*pvdA*Δ*fpvA* cells (19.8% ($p < 10^{-6}$) or 24.0%

($p < 10^{-6}$)) in the presence of photon (Fig. 2b and Supplementary Movie 6) or antimicrobial stress (Supplementary Fig. 2b). These findings demonstrate that the accumulation of PVDI in bacterial periplasms enables *P. aeruginosa* cells to survive in the presence of different environmental stresses.

**Attenuation the efflux of PVDI for private use**. We observed that the concentration of PVDI in the supernatant of bacterial cultures (wild type) negatively correlated with the concentration of tobramycin (Fig. 3a) when we used low dose tobramycin ($\leq 1.0$ μg mL$^{-1}$) to treat bacteria. Note that the tobramycin concentration applied in this experiment did not affect the growth of cells (Supplementary Fig. 1). By contrast, the concentration of PVDI accumulated in cells positively correlated with the concentration of tobramycin (Fig. 1e). These findings suggest that *P. aeruginosa* cells prefer reserving PVDI molecules intracellularly rather than secreting these molecules in the presence of antimicrobial stress. To quantify this phenomenon, we used the

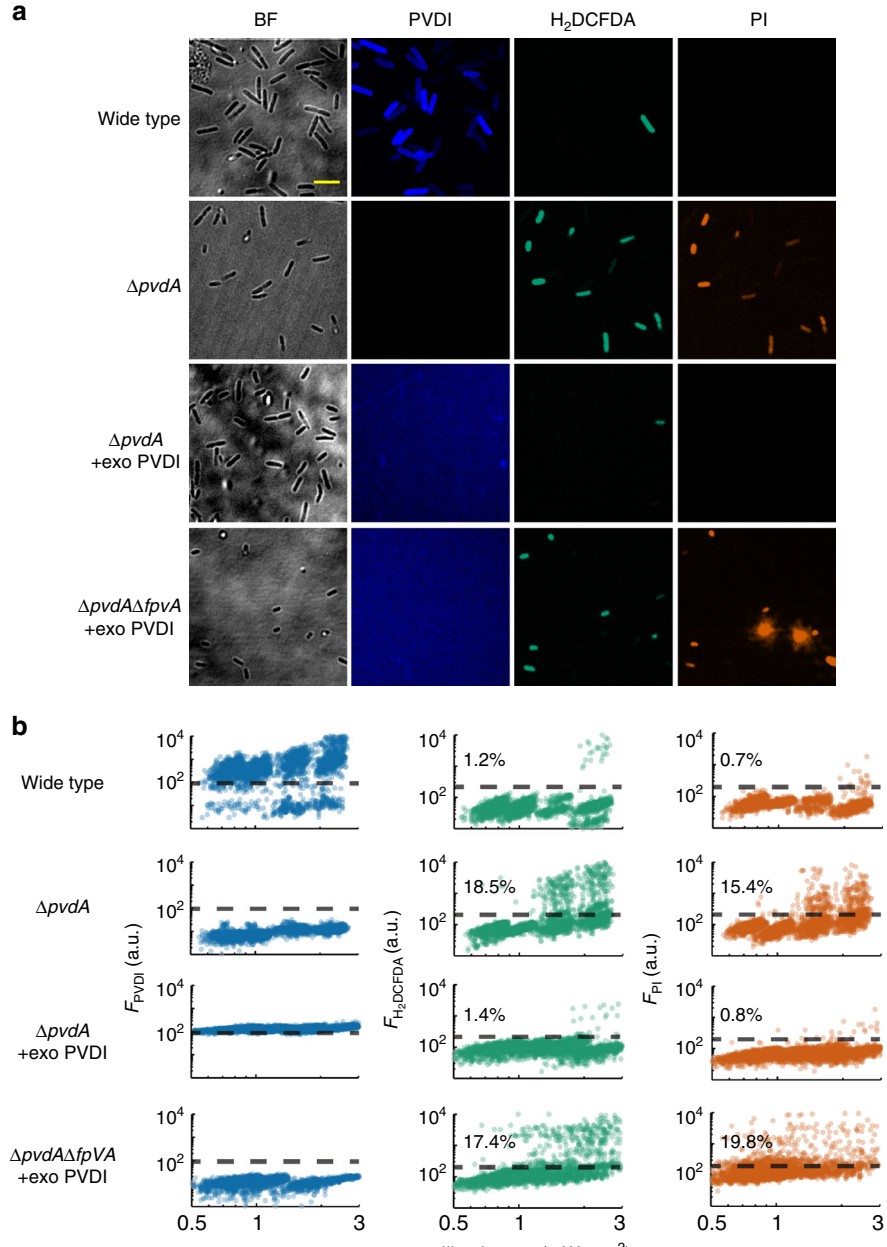

**Fig. 2** Accumulation of pyoverdine in bacteria allows them to survive in the presence of environmental stresses. **a** Representative bright-field (gray) + multi-color confocal images show that the accumulation or the exogenous addition of PVDI (5.0 μM) enables wide-type *P. aeruginosa* or Δ*pvdA* mutant to survive in the presence of a stronger photon-stress (3.00 mW cm$^{-2}$), where blue, bluish green or vermillion colors represents PVDI, H$_2$DCFDA or propidium iodide (PI) fluorescent intensity respectively, Δ*pvdA* or Δ*pvdA*Δ*fpvA* represents a mutant that deficient in production of PVDI or deficient in both production of PVDI and uptake of exogenous PVDI, respectively. +exo PVDI represents the exogenous addition of 5.0 μM PVDI. **b** Photon-stress dependence of PVDI (blue), H$_2$DCFDA (bluish green) or PI fluorescence intensity (vermillion) in single cells, where the dots present data from single bacteria. Δ*pvdA* and Δ*pvdA*Δ*fpvA* + exo PVDI groups show significantly higher percentage than that of the wide type group, both for H$_2$DCFDA channel and PI channel (one-way RM ANOVA versus wide type data, $p < 10^{-6}$ for Δ*pvdA* and Δ*pvdA*Δ*fpvA* + exo PVDI). Δ*pvdA* + exo PVDI group shows no significantly difference with wide type group, both for H$_2$DCFDA channel ($p = 0.8054$) and PI channel ($p = 0.2557$). The black dashed line represents an intensity threshold to distinguish whether the fluorescence intensities arising from single cells is significantly greater than the average. Single cells were exposed to light stimulations for 5 hours. Scale bar for all images are 4 μm

following rate equation to describe the kinetics of the intracellular concentration of PVDI ($[\text{PVDI}]^i$):

$$\frac{d[\text{PVDI}]^i}{dt} = P + \alpha[\text{PVDI}]^o - \gamma[\text{PVDI}]^i - \mu[\text{PVDI}]^i, \quad (1)$$

where $[\text{PVDI}]^o$ is the extracellular concentration of PVDI, $P$ is the productivity of PVDI, $a$ and $\gamma$ are the uptake and efflux rates of

extracellular and intracellular PVDI, respectively, and $\mu$ is the growth rate of a cell. Note that (1) exponential growth dilutes the intracellular concentration of PVDI with the constant of $\mu$ and (2) the ratio $(\gamma/\mu)$ of the terms $\gamma[\text{PVDI}]^i$ and $\mu[\text{PVDI}]^i$ accurately quantifies how cells allocate their produced PVDI, in which $\gamma/\mu > 1$ or $\gamma/\mu < 1$ indicates that cells prefer to secrete or reserve PVDI molecules produced by themselves, respectively. To directly

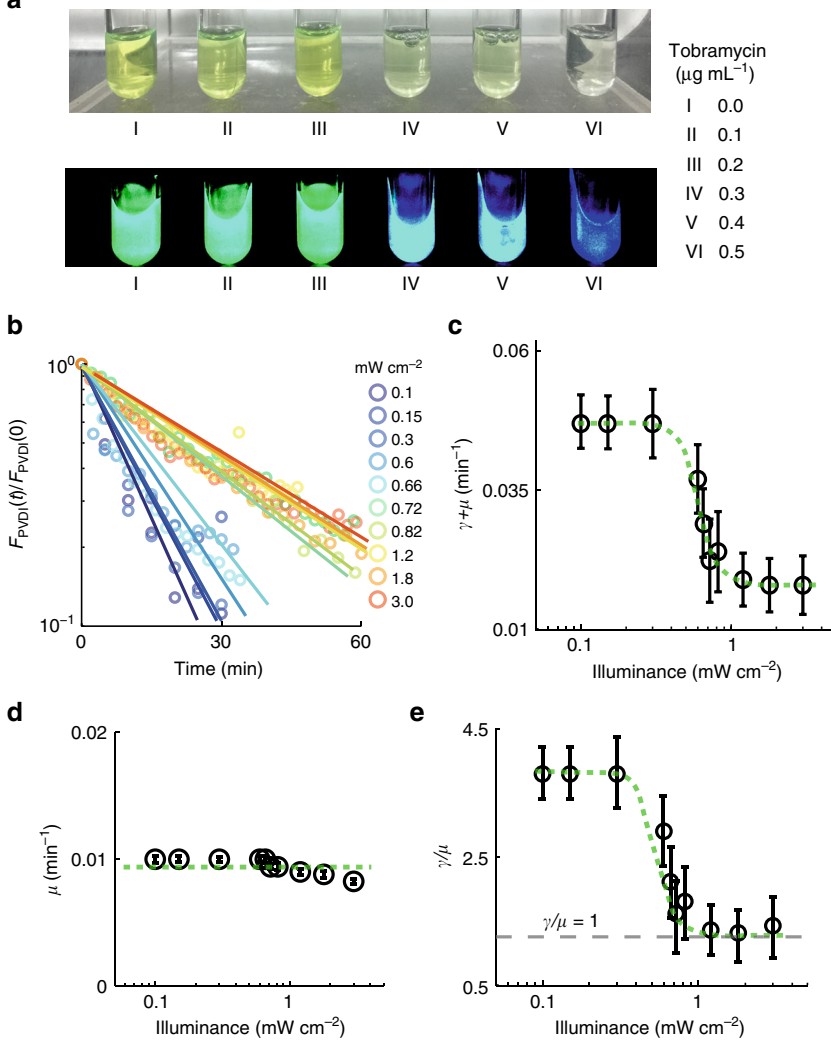

**Fig. 3** *P. aeruginosa* tunes down the efflux of PVDI in the presence of environmental stresses. **a** Representative regular + fluorescent photos show that the amount of PVDI in the supernatant of bacterial cultures negatively relates to the concentration of tobramycin, where all the bacterial cultures were harvested at an identical bacterial density (O.D.$_{600}$ = 0.3). **b** Decline of PVDI fluorescence intensity in the presence of different photon-stresses indicates that bacteria tune down the efflux of PVDI, where dots and lines represent the average intensities and the fitting lines using exponential fitting, respectively, with the magnitude of the photon-stress of 0.1, 0.15, 0.3, 0.6, 0.66, 0.72, 0.82, 1.2, 1.8, and 3.0 mW cm$^{-2}$, from bottom to top. **c–e** Photon-stress dependence of **c** ($\gamma + \mu$), **d** $\mu$, and **e** $\gamma/\mu$, where $\gamma$ or $\mu$ represents the PVDI efflux rate or bacterial growth rate, respectively. The error bars in **c–d** are the standard deviation of experimental values

measure the value of $\gamma/\mu$ in the presence of environmental stresses, we pre-cultured $\Delta pvdA$ cells ($P = 0$) in the presence of ePVDI (50 μM) to allow them to absorb sufficient PVDI; we then rapidly removed all ePVDI from bacterial cultures by washing of a fresh medium ($\alpha[\text{PVDI}]^o = 0$) and recorded the decline of $[\text{PVDI}]^i$ as well as the growth of cells in situ in the presence of different photon stresses ranging from 0.10 mW cm$^{-2}$ to 3.00 mW cm$^{-2}$. These specific boundary conditions considerably simplify the Eq. (1). In all tested conditions, the decline of $[\text{PVDI}]^i$ scaled linearly to the time elapsed ($t$) using a semilogarithmic plot (Fig. 3b), indicating that the decline of $[\text{PVDI}]^i$ in these $\Delta pvdA$ cells follows first-order kinetics: $[\text{PVDI}]^i(t)/[\text{PVDI}]^i(t = 0) = \exp[-(\gamma + \mu)t]$. The decay rate $(\gamma + \mu)$ of $[\text{PVDI}]^i$ correlated nonlinearly (Hill-like) with the magnitude of photon stresses (Fig. 3c), indicating that $\gamma + \mu$ decreased considerably only in the presence of stronger photon stresses (> 0.50 mW cm$^{-2}$). Next, we evaluated the value of $\mu$ in the presence of different photon stresses (Fig. 3d), which enabled

us to calculate directly the value of $\gamma/\mu$. Note that the illumination conditions applied in this experiment did not affect the growth of cells. Figure 3e shows that the Hill-like transition existed in the profile of $\gamma/\mu$, where the value of $\gamma/\mu$ remained at a constant of 3.70 if photon stress was relatively low (< 0.30 mW cm$^{-2}$); however, its value rapidly decreased to 1.05 if photon stress became higher (>1.00 mW cm$^{-2}$).

It has been reported that *P. aeruginosa* uses an efflux pump (PvdRT-OpmQ) composed of PvdR, PvdT, and OpmQ to secrete PVDI to the environment in a specific manner[34]. To further investigate whether the attenuation of PVDI efflux in the presence of environmental stress is caused by the tuning down of the efflux pump of PvdRT-OpmQ, we examined the response of the $\Delta pvdRT - opmQ$ mutant deficient in PvdRT-OpmQ expression to photon stress. We found that the initial level of intracellular PVDI of $\Delta pvdRT - opmQ$ mutant was markedly higher than that of wild-type cells (Supplementary Fig. 4), however, light stimulation (3.00 mW cm$^{-2}$) did not result in

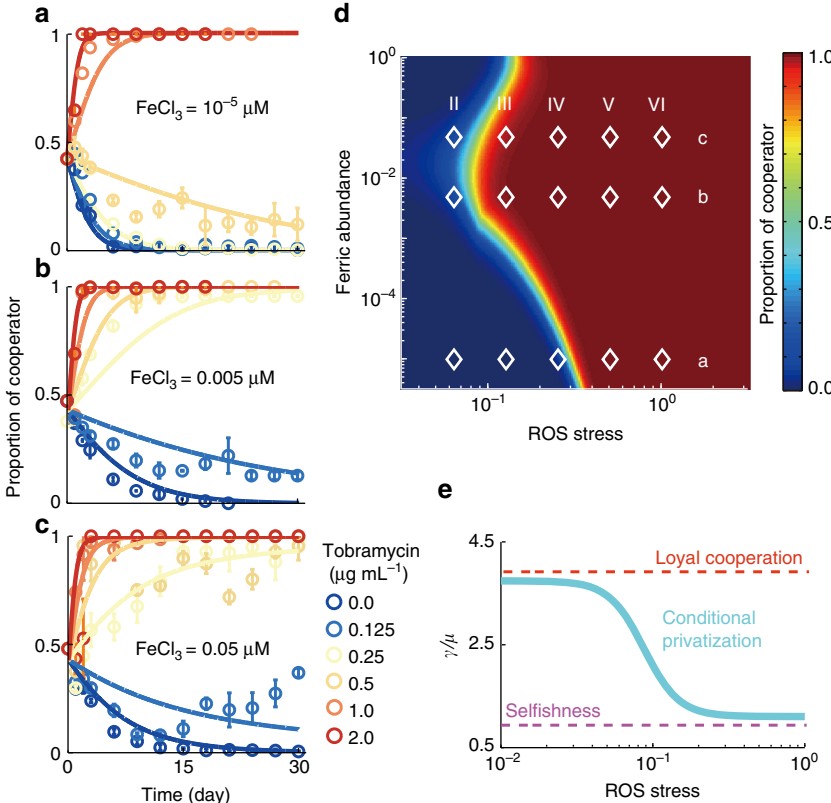

**Fig. 4** Conditional privatization stabilizes bacterial cooperation in the presence of environmental stresses. **a–c** Direct evolution of the producer (cooperator, wild-type) and non-producer strain (cheater, $\Delta pvdA$ mutant) of PVDI in the presence of different amounts of $FeCl_3$ (**a** $1 \times 10^{-5}$ μM), (**b** 0.005 μM), and (**c** 0.05 μM) and different amounts of tobramycin (0.0 to 2.0 μg mL$^{-1}$), where symbols and lines represent the experimental data and the theoretical calculations, respectively. The final concentration of tobramycin in each panel is 0.0, 0.125, 0.25 0.5, 1.0, and 2.0 μg mL$^{-1}$, from bottom to top, respectively. **d** Producer fraction as a function of the ferric abundance and the normalized ROS stress, where the normalized ROS stress = [Tobramycin]/ $IC_{50}$, the symbol [] represents the concentration, $IC_{50} = 2.0$ μg mL$^{-1}$. Colors represent the fraction of producers in the end of direct evolution, red or blue color indicates the cooperator or cheater dominated, respectively. Symbols represent the evolutionary outcomes arising from experiments in the presence different amount of ferric ions or tobramycin; i.e., $10^{-5}$ μM (**a**), 0.005 μM (**b**), 0.05 μM (**c**) $FeCl_3$, 0.125 (II), 0.25 (III), 0.5 (IV), 1.0 (V), 2.0 (VI) μg mL$^{-1}$ tobramycin. **e** Schematic showing the principle response curves of the efflux of the public goods in the presence of various environmental stresses, where the curve with cyan, red or magenta color represents the conditional privatization, loyal cooperation or selfishness respectively. The error bars in **a–c** are the standard deviation of experimental values

further accumulation of PVDI in $\Delta pvdRT - opmQ$ cells. This finding strongly suggests that *P. aeruginosa* can specifically tune down the efflux pump of PvdRT-OpmQ in the presence of environmental stress. Hence, we conclude that an existing molecular mechanism enables *P. aeruginosa* cells to tune down the efflux pump of PvdRT-OpmQ to reserve public PVDI for private use in the presence of environmental stress. We term this mechanism 'conditional privatization'.

**Resistance of cheater invasion via conditional privatization.** Griffin et al.[23] reported that the cheating strain of *P. aeruginosa* (which does not secrete PVDI) eventually outcompetes the cooperative strain (wild type) in various iron-limited conditions, thus resulting in the collapse of cooperation. This collapse occurs because the cheater does not produce the metabolically expensive PVDI, but instead exploits the public goods secreted by cooperators. We predicted that conditional privatization enables the cooperative stain of *P. aeruginosa* to withstand invasion by cheaters in the presence of environmental stresses. To test this prediction, we mixed an equal ratio of cheater $\Delta pvdA$ and cooperator (wild type) strains and cocultured them for 30 days to allow them to evolve in the presence of environmental stresses. We adjusted the magnitude of environmental stresses by adding

tobramycin to achieve various final concentrations (0.0 to 2.0 μg mL$^{-1}$) in the culture medium, with the ferric concentration ranging from $1.0 \times 10^{-5}$ to $5.0 \times 10^{-2}$ μM. We observed that under all iron-limited conditions, the $\Delta pvdA$ strain outcompeted the wild-type strain after 15 days in the absence of a stronger environmental stress ([tobramycin] $\leq 0.125$ μg mL$^{-1}$) (Fig. 4a–c); this finding is consistent with that reported in a previous study[23]. By contrast, the wild-type strain outcompeted the $\Delta pvdA$ strain after 7 days in the presence of a stronger environmental stress ($1.0 \leq$ [tobramycin] $\leq 2.0$ μg mL$^{-1}$). This result indicates that the presence of environmental stresses prevent invasion by the cheater strain. The protective activity of intracellular PVDI plays an essential role in the survival of the cooperator strain in the presence of environmental stress; however, exogenously added PVDI can protect the cheater strain $\Delta pvdA$ in the presence of environmental stress (Fig. 2, Supplementary Fig. 2). This finding indicates that public PVDI can be exploited by cheaters both to acquire iron in iron-limited conditions and protect themselves in the presence of environmental stress. We have summarized and plotted the results of our evolutionary experiments in Fig. 4d. This figure shows that as the environmental stresses intensify, the final evolutionary outcome changes, with the dominance of the cheater changing to that of the cooperator.

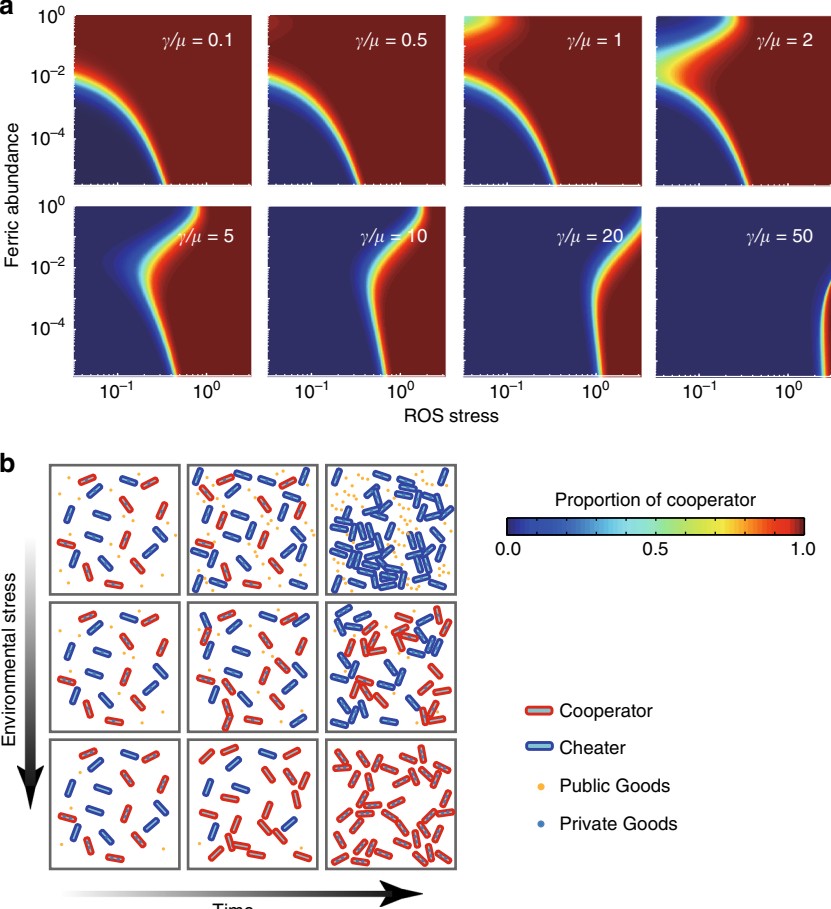

**Fig. 5** Evolutionary outcomes dominated by different strategies. **a** Theoretical calculations show $\gamma/\mu$ dependence of producer fractions as a function of the abundance of ferric ions and the normalized environmental stress, where $\gamma/\mu$ ranges from selfishness ($\gamma/\mu = 0.1$) to loyal cooperation ($\gamma/\mu = 50$). **b** Schematic showing that conditional privatization stabilizes bacterial cooperation in the presence of environmental stresses

To determine the role played by conditional privatization in our evolutionary experiments, we built a theoretical model to link environmental stresses to evolutionary outputs by considering the following factors: (1) production of public goods (PVDI) enables the growth of all populations in iron-limited conditions, but this is metabolically expensive and thus reduces the growth rate of producers (Eqs. 2 and 3 in the Methods); (2) conditional privatization enables producers to reserve those public molecules for private use in the presence of environmental stresses, as indicated by the change from $\gamma/\mu > 1$ to $\gamma/\mu < 1$ when the magnitude of environmental stresses exceeds a certain threshold (Eq. 4 in the Methods); and (3) private use of public goods enables producers to survive in the presence of environmental stresses (Eq. 7 in the Methods). Note that our modeling based on solving the differential equations (Eqs. 8–12), but alternative modeling based on the adaptive dynamics framework also could be applied to determine the evolutionary role played by conditional privatization[35,36]. To directly compare evolutionary outputs arising from conditional privatization, we assumed that producers may employ distinctive strategies in response to environmental stresses (Fig. 4e), including (1) "loyal cooperation" through which a bacterium always secretes public goods despite environmental stresses, as indicated by a maintained constant of $\gamma/\mu = \text{cons.} \gg 1$ (Eq. 4 in the Methods) and (2) "selfishness" through which a bacterium does not secrete public goods, as indicated by a maintained constant of $\gamma/\mu = \text{cons.} \ll 1$ (Eq. 6 in the Methods). Theoretical calculations indicate that our modeling

can reproduce evolutionary kinetics (as indicated by solid lines in Fig. 4a–c) and final evolutionary outcomes (as indicated by the color map in Fig. 4d) that are fully consistent with our experimental observations (as indicated by symbols in Fig. 4a–c or d) in all tested conditions. By contrast, our model predicts that bacterial cooperation would eventually collapse if a bacterium always secreted public goods despite environmental stresses (Fig. 5a). After comparing the evolutionary outcomes dominated by different strategies, including conditional privatization (Fig. 4d) and loyal cooperation or selfishness (Fig. 5a), we concluded that conditional privatization in the presence of environmental stresses is a novel mechanism that can prevent the "tragedy of the commons".

**Competition in wild-type and $\Delta pvdRT$—$opmQ$ strain.** First, we examined the growth rate of $\Delta pvdRT - opmQ$ strain in different growth conditions to evaluate the cost of privatization of public PVDI. We found that (1) in the iron-limited condition of $FeCl_3 = 1.0 \times 10^{-5}\,\mu M$ (Supplementary Fig. 5) or $FeCl_3 = 0.1\,\mu M$, the growth rate of ($\Delta pvdRT - opmQ$) mutant ($\mu_3 = 0.0032\,min^{-1}$ at $FeCl_3 = 1.0 \times 10^{-5}\mu M$, Supplementary Fig. 8b or $\mu_3 = 0.0035\,min^{-1}$ at $FeCl_3 = 0.1\,\mu M$, Supplementary Fig. 9b) was slower than that of the wild-type strain ($\mu_1 = 0.0039\,min^{-1}$ at $FeCl_3 = 1.0 \times 10^{-5}\mu M$, Supplementary Fig. 8a or, $\mu_1 = 0.0099\,min^{-1}$ at $FeCl_3 = 0.1\,\mu M$, Supplementary Fig. 9a); and (2) the exogenous addition of PVDI (ePVDI = $5.0\,\mu M$) markedly increased the growth rate of the $\Delta pvdRT - opmQ$ strain ($\mu_3 =$

0.0100 min$^{-1}$ at FeCl$_3$ = 0.1 µM + ePVDI = 5.0 µM, Supplementary Fig. 9c) in iron-limited conditions. These results arising from $\Delta pvdRT - opmQ$ strain (constitutive privatization) indicated that (1) production of PVDI is metabolically expensive whether produced PVDI is public or private; (2) the secretion of private PVDI is beneficial to bacterial growth in the iron-limited condition; and (3) PVDI constitutive privatization strain can exploit public PVDI to benefit in iron-limited conditions.

Next, we predict the evolutionary outcome of competition between the wild-type (conditional privatization) and $\Delta pvdRT - opmQ$ (constitutive privatization) strains using our theoretical model (see Methods). Our calculation indicated that (1) $\Delta pvdRT - opmQ$ coexists with the wild-type strain in the presence of ROS stress (Supplementary Fig. 10a); and (2) the relative yield is negative in the coexistence region (Supplementary Fig. 10b). These results are in contrast with those of the competition of $\Delta pvdA$ and wild-type strain, in which the relative yield is positive in the region dominated by $\Delta pvdA$ cells (Supplementary Fig. 10c, d). There the results indicated that the $\Delta pvdRT - opmQ$ strain is not able to invade the wild-type strain in the presence of ROS stress. To further test this, we predicted the evolutionary outcomes of competition among the wild-type (conditional privatization), $\Delta pvdRT - opmQ$ (constitutive privatization) and $\Delta pvdA$ (cheater strain) strains. Supplementary Fig. 11a and b show that the wild-type strain is favored in the presence of ROS stress; that is, the wild-type strain cannot be invaded by either $\Delta pvdRT - opmQ$ or $\Delta pvdA$ strain in the presence of ROS stress.

**Roles played by private PVDI**. To determine the roles played by private PVDI in our evolutionary experiments, we directly calculated the kinetics of the extracellular PVDI concentration ([PVDI]$^o$(t)), the growth rate ($\mu_1(t)$ or $\mu_2(t)$), death rate ($p_1(t)$ or $p_2(t)$), and intracellular PVDI concentration ([PVDI]$^i_1$(t) or [PVDI]$^i_2$(t)) of the cooperator or cheater strain during competition by using our theoretical model. Our calculations indicated that (1) the cooperator strain outcompetes the cheater strain in the presence of environmental stress (i.e., FeCl$_3$ = $5.0 \times 10^{-3}$µM and Tobramycin = 2.0 µg mL$^{-1}$, Supplementary Fig. 12a); (2) the domination of the cooperator strain led to the accumulation of extracellular PVDI, resulting in a gradual increase of [PVDI]$^o$ over time (Supplementary Fig. 12b); (3) an increase of [PVDI]$^o$ led to an increase in the growth rate of both the cooperator and cheater strains (Supplementary Fig. 12c, note that $\mu_2(t)>\mu_1(t)$ despite their increase); (4) the intracellular PVDI concentration was high in the cooperator strain (Supplementary Fig. 12e) because ROS stress triggers the privatization of PVDI, which further results in the reduction of the death rate of cooperator strain; and (5) the intracellular PVDI concentration in the cheater strain remains at a low level in the initial stage and then gradually increases with increase in [PVDI]$^o$ (note that the cheater strain could also tune down the efflux of PVDI in the presence of environmental stress), which results in an increase in the death rate of the cheater strain in the initial stage, which gradually decreases in the later stage. These results indicated that the wild-type strain outcompetes those non-producers in the presence of ROS stress due to protective activity of intracellular PVDI. To further test this interpretation, we predicted the evolutionary outcomes of competition between the wild-type strain (conditional privatization) and $\Delta pvdA$ strain (cheaters) with an additional assumption that intracellular PVDI does not enable cells to survive in the presence of ROS stress. We found that under this assumption the cheater always outcompetes the cooperator strain despite ROS stress (Supplementary Fig. 13).

## Discussion

We have shown that when *P. aeruginosa* is exposed to photon and antibiotic stresses, it suppresses secretion of the siderophore PVDI and instead reserves PVDI intracellularly. The intracellular PVDI protects cells from ROS damage and increases cell survival in stressful conditions. We term this strategy 'conditional privatization': the bacteria secrete public goods to cooperate with other cells in the absence of environmental stresses, whereas they reserve those public goods for private use once the environment becomes stressful. This conditional strategy outcompetes the non-producing, cheater strategy under environmentally stressful circumstances (Fig. 5b). The molecular mechanism of conditional privatization follows a hierarchical procedure: (1) environmental stresses generally trigger ROS generation in *P. aeruginosa* cells; (2) ROS triggers cells to tune down the efflux pump of PvdRT − OpmQ to decrease the secretion of PDVI, leading to the accumulation of PDVI in cells; and (3) the accumulation of PVDI enables cells to survive in the presence of environmental stresses, presumably through the repression of hydroxyl radical formation. However, the mechanism through which the efflux pump of PvdRT − OpmQ is tuned down in response to ROS remains elusive, meriting a more detailed investigation.

The private use of public goods by their producer might be a common mechanism that enables the persistence of cooperative behaviors in microbes. For example, Gore et al.[37] reported that the budding yeast *S. cerevisiae* can retain approximately 1% of public goods for private use, and that such a seemingly insignificant level of privatization allows cooperative cells to invade a population of cheaters[37,38]. In addition, our findings highlight that environmental stresses affect cooperation in microbes, which may be compared to the effect of resource scarcity on cooperation in human society. For example, Gatiso et al.[39] found that people who encountered resource scarcity were less cooperative compared with those who had more abundant resources in an investigation of the extraction of trees from a communally managed forest in Ethiopia. It should be emphasized that conditional strategies, including conditional cooperation or cheating, are quite general. For example, Rustagi et al.[40,41] reported that conditional cooperation and costly norm enforcement can stabilize large-scale cooperation for common management. By contrast, our findings highlighted that one conditional strategy can coordinate the secretion of PVDI that plays the dual function; i.e., public PVDI for iron scavenging or private PVDI against ROS stress.

As a crucial opportunistic human pathogen, understanding the cooperation of *P. aeruginosa* is of great significance to clinical therapies[42]. Exploitation of iron in the hosts of *P. aeruginosa* depends on the cooperative production of siderophores[43] and virulence correlates with siderophore production[44]. Therefore, our findings of how *P. aeruginosa* alters siderophore secretion in the presence of environmental stresses can not only advance our understanding of the maintenance of cooperation but also provide an opportunity to deepen our understanding of the relationship between cooperation and virulence. This advance may assist in the rational development of novel therapies for pathogenic populations[45].

## Methods

**Bacterial strains and growth conditions**. Bacterial strains and plasmids used in this study are listed in Supplementary Table 1. Strains were grown on LB agar plates at 37 °C for 12 h. Monoclonal colonies were inoculated and cultured with a minimal medium (SSM) at 37 °C under an aerobic condition, in which the medium had the following composition per liter: K$_2$HPO$_4$·3H$_2$O, 7.86 g; KH$_2$PO$_4$, 3.00 g; (NH$_4$)$_2$SO$_4$, 1.00 g; MgSO$_4$·7H$_2$O, 0.20 g; succinic acid, 4.00 g NaOH was added to the medium to adjust pH = 7.0 before sterilization. Different amounts of FeCl$_3$ aqueous solution (pH = 2.0) were added to the medium with the resultant concentrations of 10$^{-5}$, 0.005, 0.05, 0.1, or 100 µM to generate iron-limited (FeCl$_3$ =

$10^{-5}$ to 0.1 μM) or iron-rich conditions ($FeCl_3$ = 100 μM). Iron-rich or iron-limited conditions were also generated using the Casamino Acid medium (CAA) or CAA medium + 20 mM apotransferrin (Sigma-Aldrich), in which one liter CAA medium consists of Casamino Acid (BBI Life Sciences), 5.00 g; $K_2HPO_4 \cdot 3H_2O$, 1.18 g; $MgSO_4 \cdot 7H_2O$, 0.25 g. Different amounts of tobramycin (0.0 to 4.0 μg mL$^{-1}$) or gentamicin (2.0 μg mL$^{-1}$) were added in the medium to adjust the antimicrobial stresses. We found that wild-type *P. aeruginosa* bacteria grow slowly in the CAA medium +20 mM apotransferrin during the first 12 h, reflecting iron limitation. However, growth became fast after 20 h of culturing and the bacterial culture eventually reached high density (OD = 1.0, 44 h), suggesting that iron abundance is no longer limited. This is presumably because *P. aeruginosa* can gradually secrete proteases that may degrade the apotransferrin[46], thus releasing these chelating irons in the later stage. Most importantly, we found that PVDI cannot protect cells against ROS stress at a later culturing stage (24 h) (Supplementary Fig. 3g, h) in the CAA medium +20 mM apotransferrin, which would further result in a distinctive evolutionary outcome (Supplementary Fig. 13), in which the PVDI non-producer outcompetes the producer strain in the presence of antimicrobial stress[47]. Our results indicated that the CAA medium + 20 mM apotransferrin is not reliable in generating an iron-limited condition. Therefore, the SSM medium was used in our evolutionary experiments.

### Construction of the *P. aeruginosa* mutants.

Deletion mutants Δ*pvdA*, Δ*pvdA*Δ*fpvA*, Δ*pvdRT* − *opmQ* were constructed by allelic exchange using a modified procedures for *P. aeruginosa*. We constructed unmarked deletion mutants by Flp-mediated excision of the antibiotic resistance marker[48]. Firstly, two DNA fragments, each obtained from upstream and downstream of target deletion genes by PCR, were cloned into a gene replacement vector pEX18Ap via a three-piece ligation. Then the constructed plasmid was inserted into a gentamicin/tetracycline resistant piece between upstream and downstream fragment. The final plasmid was electroporated into PAO1 and the corresponding recombinant strain was identified by screening on LB agar plates containing 5% (w/v) sucrose within 30.0 μg mL$^{-1}$ gentamicin. Then the strains were electroporated with a pFLP2 plasmid and distinguished on LB agar plates containing 5% (w/v) sucrose for the excision of the resistance marker. Mutations were finally verified by PCR and sequencing. The constructed plasmids are listed in Supplementary Table 1. The PAO1 psfGFP, PAO1 pmCherry, PAO1 Δ*pvdA* psfGFP strains were constructed by specific insertion into PAO1 genome with the plasmid pUC18T-mini-Tn7T-Gm[49]. Firstly, target fluorescent protein piece was amplified by PCR and cloned into the plasmid pUC18T-mini-Tn7T-Gm. Then the plasmid carrying sfGFP/mCherry, was electroporated into PAO1 or PAO1 Δ*pvdA* with a helper plasmid pTnS2 and finally identified on LB agar plates containing 30.0 μg mL$^{-1}$ gentamicin.

### Pyoverdine extraction and purification.

Pyoverdine extraction and purification were exactly followed as previously reported protocol[50]. Briefly, 1 litre conical flasks, which contained 200 mL SSM, was inoculated with PAO1. The bacteria were grown to exponential-phase under an aerobic condition at 37 °C. After that, the bacterial culture were centrifuged at 10,000×g for 10 min at 4 °C. The supernatant was further filtered by a 0.22 μm filter (Merck Millipore). The cell-free supernatant was resuspended in 1 M HEPES buffer (pH = 7.0) and then applied to a Chelating Sepharose Fast Flow column (1 by 5 cm; GE Healthcare, eluent flow rate = 300 mL h$^{-1}$), where the column was pre-saturated with $CuSO_4$ and equilibrated with 20 mM HEPES buffer (pH 7.0) containing 100 mM NaCl. The column was then washed with 20 mM HEPES buffer and eluted with 20 mM acetate buffer (pH = 5.0) containing 100 mM NaCl. Fractions (5 mL) were collected and the absorbance at 400 nm ($A_{400}$) of which was determined with a spectrophotometer (NanoDrop 2000, Thermo SCIENTIFIC). The pyoverdine-containing fractions were separately pooled and lyophilized. Each fraction of dried pyoverdine was dissolved with 1 mL 10 mM EDTA and then applied to a Sephadex G-15 column (1 by 80 cm; GE Healthcare) which wax pre-equilibrated with deionized water. The column was eluted with ultrapure water at a flow rate of 20 mL h$^{-1}$ and 15 fractions (4 mL) were collected and monitored the absorbance of UV–Vis spectrum. The fractions with the highest absorbance at 385 nm were chosen as purified-pyoverdine and then were pooled, lyophilized and stored at −20 °C.

### Observation of pyoverdine productions in vivo.

Bacterial strains were inoculated into a flow cell (Denmark Technical University) and continuously cultured at 30.0 ± 0.1 °C by flowing SSM medium (3.0 mL h$^{-1}$)[51] in the presence of different environmental stresses: illumination by 405 nm laser with different powers and exposure times was used to generate photon stresses with a controllable magnitude (0.1–10.0 mW cm$^{-2}$) and different amounts of tobramycin (0.0–4.0 μg mL$^{-1}$) or gentamicin (2.0 μg mL$^{-1}$) were added to the medium to generate antimicrobial stresses. The illuminances were determined using a laser power meter (Newport 842-PE). Meanwhile, a spinning-disk confocal microscope (Revolution Andor) equipped with a 100 × oil objective and an EMCCD (iXon 897i Andor) was applied to continuously record the bright-field and confocal images. The dyes of propidium iodide (PI, Sigma) with a resultant concentration of 1.0 μg mL$^{-1}$ and 2′,7′-dichlorofluorescin diacetate (H$_2$DCFDA, Thermo Fisher Scientific) with a resultant concentration of 20 μM were pre-added in the medium. The pyoverdine, H$_2$DCFDA or PI was excited using a 405, 488, or 561 nm laser, respectively and

imaged with three different emission channels (450, 524, or 607 nm). The confocal images with different channels were further analyzed to quantify the fluorescent intensities of single cells using a standard algorithm coded by Matlab, in which ≥2000 cells were typically counted for each condition.

### Quantification of pyoverdine efflux rate under stresses.

The strain (PAO1-Δ*pvdA*) was inoculated into the channel of flow cell and pre-cultured for 1 h by flowing (3.0 mL h$^{-1}$) the SSM medium in the presence of 50.0 μM purified PVDI. Then, a flash SSM medium in the absence of PVDI was used to wash surface-attached cells for 30 min to remove PVDI presented in the environment. Afterwards, the illuminations of 405 nm laser with different powers and exposure times were used to generate different photon stresses. Meanwhile, a spinning-disk confocal microscope (Revolution Andor) equipped with a 100 × oil objective and an EMCCD (iXon 897i Andor) was applied to continuously record the bright-field and confocal images, in which PVDI was excited using a 405 nm and imaged with 450 nm emission channel. The confocal images or bright-field images were further analyzed to quantify the decline of PVDI fluorescent intensities detected in the bacterial periplasm or the growth rate of single cells, respectively, in the presence of different photon stresses.

### Direct evolution of cooperator and cheater stains.

The cooperator (PAO1 pmCherry) and cheater (PAO1 Δ*pvdA* psfGFP) strains were inoculated into SSM mediums at a 1:1 ratio in the presence of different amounts of $FeCl_3$ ($10^{-5}$, 0.005, 0.05 μM) or tobramycin (0.0, 0.125, 0.25, 0.5, 1.0, and 2.0 μg mL$^{-1}$). The control group was set by inoculating a 1:1 ratio of strains PAO1 pmCherry and PAO1 psfGFP in the SSM medium in the absence of $FeCl_3$ and tobramycin. Each condition was replicated three times. The resultant mixed bacterial cultures were further cultured in a shaker at 37 °C for 24 h. Afterwards, the bacterial cultures were diluted (v/v 1:1000) in fresh SSM mediums containing corresponding amounts of $FeCl_3$ and tobramycin for the next round of culturing. Thirty rounds of culturing were performed in total. At the end of each round, a bacterial counting sample was carefully prepared by sandwiching 2.0 μL bacterial culture between a coverslip and 2.0 mm thick agarose gel. Then, a spinning-disk confocal microscope (Revolution Andor) equipped with a 100 × oil objective and an EMCCD (iXon 897i Andor) was applied to directly image and count GFP or RFP-labeled bacteria, in which ≥10$^4$ cells were typically counted for each round.

### Theoretical modeling and evolutionary analysis.

In our model, we assume that the growth rate ($\mu_1$ or $\mu_2$) of producer (cooperator) or non-producer strains (cheater) of PVDI relates to the ferric abundance in the environments ([$Fe^{3+}$]) and the extracellular concentration of PVDI ([PVDI]$^o$) as below,

$$\mu_1 = \mu_1^{max} - \left(\mu_1^{max} - \mu_1^{min}\right) \exp\left(-a_1 [Fe^{3+}] - a_2 [Fe^{3+}][PVDI]^o\right), \quad (2)$$

$$\mu_2 = \mu_2^{max} - \left(\mu_2^{max} - \mu_2^{min}\right) \exp\left(-a_1 [Fe^{3+}] - a_2 [Fe^{3+}][PVDI]^o\right), \quad (3)$$

where the parameters ($\mu_1^{max}$, $\mu_1^{min}$, $\mu_2^{max}$ and $\mu_2^{min}$) are set by $\mu_1^{max}$ = 0.0160 min$^{-1}$, $\mu_1^{min}$ = 0.0039 min$^{-1}$, $\mu_2^{max}$ = 0.0163 min$^{-1}$ and $\mu_2^{min}$ = 0.0042 min$^{-1}$ (Supplementary Figs. 6, 7) according to the measurement of the actual growth rate of these strains in the iron-limited condition ([$Fe^{3+}$] = 1.0 × $10^{-5}$ μM) or in the presence of sufficient [$Fe^{3+}$] ([$Fe^{3+}$] = 1.0 μM) and PVDI ([PVDI] = 5.0 μM), the exponent of $a_1[Fe^{3+}]$ or $a_2[Fe^{3+}][PVDI]^o$ in the Eqs. (2) and (3) reflects directly or indirectly uptake of the ferric iron. Note that $\mu_2^{max} > \mu_1^{max}$ and $\mu_2^{min} > \mu_1^{min}$, indicating that production of PVDI is metabolically expensive and thus reduces the growth rate of producers. We also assume that the ratio of PVDI efflux rate ($\gamma$) and bacterial growth rate ($\mu$) for both strains ($\gamma_1/\mu_1$ or $\gamma_2/\mu_2$) is equal and follow the same equations in response to environmental stresses as below,

$$\text{Conditional privatization} \quad \frac{\gamma_1}{\mu_1} = \frac{\gamma_2}{\mu_2} = b_4 \cdot \left[\frac{1}{1 + b_1 \left(\frac{\text{Ens}}{E_{50}}\right)^{b_2}} + b_3\right], \quad (4)$$

$$\text{Loyal cooperation} \quad \frac{\gamma_1}{\mu_1} = \frac{\gamma_2}{\mu_2} \gg 1, \quad (5)$$

$$\text{Selfishness} \quad \frac{\gamma_1}{\mu_1} = \frac{\gamma_2}{\mu_2} \ll 1, \quad (6)$$

where $\gamma_1/\mu_1$ or $\gamma_2/\mu_2$ nonlinearly (hill-like) relates to the magnitude of environmental stresses (Ens) if bacteria follow conditional privatization, while $\gamma_1/\mu_1$ and $\gamma_2/\mu_2$ are independent of Ens if bacteria follow loyal cooperation or selfishness. Ens is normalized by dividing by the stress level ($E_{50}$) that leads 50% of cells to die. After the normalization, the coefficients $b_1$, $b_2$, $b_3$, and $b_4$ can be obtained from the fitting of the experimental data (Fig. 3e). Lastly, we assume that the death rate ($p_1$ or $p_2$) for producers and non-producers relates to the magnitude of environmental

**Table 1 Parameters used for the theoretical modeling**

| Parameter | Value |
| --- | --- |
| $a_1$ | $4\,\mu M^{-1}$ |
| $a_2$ | $2.1 \times 10^6\,\mu M^{-2}$ |
| $b_1$ | 9250 (dimensionless) |
| $b_2$ | 3.75 (dimensionless) |
| $b_3$ | 0.42 (dimensionless) |
| $b_4$ | 2.64 (dimensionless) |
| $E_{50}$ (405 nm laser) | $8\,mW\,cm^{-2}$ |
| $E_{50}$ (tobramycin) | $2.0\,\mu g\,mL^{-1}$ |
| $v/V$ | $10^{-12}$ (dimensionless) |
| $c_1$ | 0.031 (dimensionless) |
| $c_2$ | $16.06\,\mu M^{-1}$ |
| $c_3$ | 16.55 (dimensionless) |
| $c_4$ | 0.0013 (dimensionless) |
| $[Fe^{3+}]_{min}$ | $10^{-5}\,\mu M$ |
| $\alpha$ | $0.1\,min^{-1}$ |
| $\delta$ | $0.0085\,min^{-1}$ |
| $P$ | $0.0118\,\mu M\,min^{-1}$ |

stress, intracellular concentration of PVDI and ferric concentration as below,

$$p_1 = \left\{ 1 - \exp\left[ -\frac{c_1 \frac{Ens}{E_{50}} + c_4 \ln\left(\frac{[Fe^{3+}]}{[Fe^{3+}]_{min}}\right)}{c_2 [PVDI]_1^i + c_3} \right] \right\} / \Delta t$$

$$p_2 = \left\{ 1 - \exp\left[ -\frac{c_1 \frac{Ens}{E_{50}} + c_4 \ln\left(\frac{[Fe^{3+}]}{[Fe^{3+}]_{min}}\right)}{c_2 [PVDI]_2^i + c_3} \right] \right\} / \Delta t$$

(7)

where the exponent $\frac{c_1 \frac{Ens}{E_{50}} + c_4 \ln\left(\frac{[Fe^{3+}]}{[Fe^{3+}]_{min}}\right)}{c_2 [PVDI]_1^i + c_3}$ or $\frac{c_1 \frac{Ens}{E_{50}} + c_4 \ln\left(\frac{[Fe^{3+}]}{[Fe^{3+}]_{min}}\right)}{c_2 [PVDI]_2^i + c_3}$ reflects the fact that intracellular PVDI can allow bacteria to survive in the presence of environmental stress and $Fe^{3+}$ would promote hydroxyl radical formation via Fenton reaction, $\Delta t = 1$ min.

We mix producer and non-producer strains to allow them to grow together in the presence of a given environmental stress, in which the dynamics of the system are assumed to exactly follow the differential equations as below,

$$\frac{d[PVDI]_1^i}{dt} = P + \alpha[PVDI]^o - \gamma_1[PVDI]_1^i - \mu_1[PVDI]_1^i,$$

(8)

$$\frac{d[PVDI]_2^i}{dt} = \alpha[PVDI]^o - \gamma_2[PVDI]_2^i - \mu_2[PVDI]_2^i,$$

(9)

$$\frac{d[PVDI]^o}{dt} = \phi_1 \gamma_1[PVDI]_1^i + \phi_2 \gamma_2[PVDI]_2^i - (\phi_1 + \phi_2)\alpha[PVDI]^o - \delta[PVDI]^o,$$

(10)

$$\frac{d\phi_1}{dt} = \phi_1(\mu_1 - p_1 - \delta),$$

(11)

$$\frac{d\phi_2}{dt} = \phi_2(\mu_2 - p_2 - \delta),$$

(12)

where $\phi_1$ and $\phi_2$ are the volume fractions ($\phi \equiv vN/V$) of producer and non-producer strains, respectively, $v$ and $V$ are the bacterial and the system volumes, respectively, $N$ is the bacterial number, and $\delta$ is the global dilution rate in our evolutionary experiment. $\phi_1(t)$ and $\phi_2(t)$ are obtained by directly solving these differential equations using the iterative method with following initial conditions, $\phi_1(0) = \phi_2(0) = 10^{-7}$, $[PVDI]_1^i(0) = [PVDI]_2^i(0) = [PVDI]^o(0) = 0\,\mu M$. The parameters used in our model are listed in Table 1.

To improve the evolutionary analysis, we evaluated the total population size ($N(t)$) of the invading strain during the competition. Note that $N(t) \equiv \phi(t)V/v$. To ensure that $N(t)$ (or $\phi(t)$) is independent of $\delta$, we normalized $N(t)$ by $N(t)/N_r(t)$ to calculate a relative yield defined as below,

$$\text{Relative yield def} = \log_{10}[N(t)/N_r(t)], \qquad (13)$$

where $N_r(t)$ represents the total bacterial number of a reference strain (wild-type strain used for reference) grown under conditions identical to those used in the

competition assay (i.e., identical iron abundance, ROS stress and initial bacterial number). As a result, the relative yield can be used to assess whether the tested strain can invade the wild-type strain; i.e., positive or negative relative yield ($\log_{10}[N(t)/N_r(t)]>0$ or $\log_{10}[N(t)/N_r(t)]<0$) represents that the tested strain can or cannot invade, respectively.

**Data availability**. All data generated or analysed during this study are available from the authors on reasonable request.

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

## Acknowledgements

This work was supported by the National Natural Science Foundation of China (21474098), (21274141), (21522406) and the Fundamental Research Funds for the Central Universities (WK2340000066), (WK2030020023).

## Author contributions

F.J conceived the project. JH.L, ZY.J performed the experiments. ZY.J performed theoretical calculations. ZY.J, AG.X analyzed data. L.N found that the photon-stresses can trigger the accumulation of PVDI. RR.Z helped to construct bacterial strains. ZY.J and F.J contributed jointly to data interpretation and manuscript preparation. All authors reviewed the manuscript.

## Additional information

**Competing interests:** The authors declare no competing interests.

