## [Peer Review File · Nature Communications]

Reviewers' comments:

Reviewer #1 (Remarks to the Author):

Review of "Conditional privatization stabilizes bacterial cooperation"

This is a very interesting study demonstrating that cells under environmental stress retain the siderophore, pyoverdinin, at the cell surface, converting pyoverdinin from a normally cooperative factor to a 'privatized' factor under these conditions. The results are relevant and interesting, because they provide new insight into cooperation and how cooperative traits are maintained. The results also provide potential explanations of findings from this and other microbial cooperation systems, which will no doubt provide the basis of much future work.

The studies are convincing, presented in a logical order, and for the most part, clear to understand. Most potential issues were addressed. There were only a few minor suggestions:

1. Fig. 1, and the description of Fig. 1 on lines 61-76, was a bit confusing. The line "Fluorescent PVDIs were also excited when illuminated by the 405 nm laser source" (lines 67-68) seems to imply 405 nm illumination AND another approach were used, but the other approach is not described (or perhaps something obvious is being overlooked?). The line "The accumulation of PVDI in the bacterial periplasm" (lines 73-74) should be clarified by adding "under illumination conditions $>3.00 \text{ mW cm}^{-2}$ ". Also if possible, results with a *pvdA* mutant should be added to Fig. 1 to demonstrate that the method described specifically detects PVDI and not something else. Although such results are presented in Fig. 2, the manuscript would read more clearly if they were also provided in Fig. 1. Also, at what time point were the images and data in Fig. 1c, d and e collected? Presumably, cells from all the compared conditions were taken at the same time but this is not indicated.
2. Fig. 2, as in Fig. 1, the time of illumination needs to be indicated.
3. Fig. 3 legend, line 457, "bacteria can tune down the efflux of PVDI" should be clarified with "in correlation with increasing stress." Also, it is not known that efflux is specifically the mechanism of PVDI release from cells.
4. Fig. 4D – the symbols are missing from the figure legend or not indicated.. It is not clear what the numerical numbering in the illustration represents.
5. It seems, based on Fig. 4 and the description of the model, there is a mistake in the figure legend. The range of γ/μ is from 0.1 (selfishness) to 50 (loyal cooperation), not 0.1 (loyal cooperation) to 50 (selfishness) as stated.
5. Fig. 5 – the colors in part B should be changed so they match part A.
6. Fig. 5 and lines 192-203. This is very interesting. Why is it that environmental stress

seems to result in a greater proportion of cheaters in the models, is this simply due to fitness effects?

Reviewer #2 (Remarks to the Author):

Review "Conditional privatization stabilizes bacterial cooperation".

The paper "Conditional privatization stabilizes bacterial cooperation" investigates the public goods dilemma arising from siderophore secretion in colonies of *Pseudomonas aeruginosa*. It describes a mechanism by which through reactive oxygen species stress (ROS), the iron-scavenging siderophore pyoverdine (PVDI) becomes "private", and therefore unexploitable by non-secreting cheaters. The authors conclude that this mechanism that they call "conditional privatization" may be a way to stabilize otherwise fragile cooperative traits.

The paper is written in quite a clear way, and the figures are nice and communicate the results well. First, the authors show that laser excitation and low doses of the antibiotic tobramycin both induce cellular stress through ROS, and that periplasmic accumulation of PVDI can protect cells. The authors then conduct competition assays and employ a mathematical model to show that stress-induced cessation of PVDI secretion---conditional privatization---can allow PVDI producers to outcompete cheaters where constitutive secretors (here called 'loyal cooperators') would not.

I think the paper shows a potentially neat mechanism by which PVDI secretion may be inherently protected from exploitation. But to be published, some critical additional data is required, and some of the evolutionary analyses need to be revised.

The authors refer to the observation that stress tends to favor competitive phenotypes, and contrast this with their findings that stress here appears to stabilize the cooperative trait of PVDI secretion through privatization. This is no contradiction as privatizing a cooperative public goods trait as a result of stress appears competitive to me.

To investigate the effect of stress on PVDI secretion and compare fitness between secretors and cheaters, the authors frequently switch between ROS-stress induced by laser and by antibiotic treatment. For example, figure 3 shows that supernatant of bacterial cultures containing increasing concentrations of tobramycin had decreasing amounts of PVDI, and that increased laser intensities led to higher PVDI retention. But while the authors do show that illumination conditions did not affect the growth of cells, such an analysis is missing for the antibiotic treatment. This should be added to clarify if supernatant concentrations ought to be corrected for cell density, and for the following competition assays.

The competition assays and mathematical analyses are asking the question: does a PVDI producer (constitutive or conditionally private) end up with a higher frequency than a cheater under different iron limitations (then PVDI becomes an exploitable 'common good'), and different ROS stress levels. ROS mediated privatization of PVDI occurs at some

threshold level through an unidentified mechanism (is there evidence that this is 'active' and that transporters are downregulated, or is it simply chemical such that the chelating reaction in the periplasm prevents passage through the outer membrane?). Since privatization of PVDI here appears to have no associated cost, somewhat unsurprisingly then ROS induced privatization leads to an advantage for cooperators. This could be a two-fold benefit: the protection against ROS conferred by PVDI could mean that cheaters die more, or it could be that exploitation is avoided such that cheaters cannot grow under iron limitation. Which one it is cannot be seen from the data as only proportions of cooperators (both in model and experiment) are shown.

PVDI is costly to secrete but enables growth under iron limitation to all cells in the community; it can thus be exploited. But when made private, it is then not available to cheaters any longer and in addition protects cooperators from cell damage and death, improving cooperators relative growth rate over cheaters even when iron is abundant. This begs the question: where is the cost to privatization of the public good? Again, one would need information on total cell numbers / yields from the experiments and the model to investigate the potential cost to not secreting PVDI.

The competition assays and model results are therefore insufficient for the rather sweeping evolutionary claims made towards the end. Cooperation via public good secretion can be evolutionary stable if, for example, colonies of cooperators grow to much higher densities than colonies invaded by cheats. The former would then yield large numbers of cooperators to the 'global' count of cells, while the latter would add only small numbers of cheats. It is therefore important to have an idea of 'yield' in competition assays and mathematical models, and it is critical to not only investigate local competition for an evolutionary analysis. Especially the mathematical model clearly lends itself for more thorough evolutionary analysis. For example, one could use an adaptive dynamics framework (e.g. doi:10.1098/rspb.2009.1480, doi:10.1007/BF02409751) to study stress-induced privatization of PVDI, but again, without cost this does not seem like much of a dilemma.

Finally, I would prefer removing the discussion section on evolution in human societies, as this appears to be too much of a stretch for what is not really an evolutionary analysis. Also there are some minor issues I noticed that I am attaching as a list at the end (non-exhaustive).

I sum, I think this paper has much potential due to the thorough investigation of the PVDI phenotype as a function of ROS. To be published in Nature Communication and to justify the evolutionary claims, however, requires more work, if at least via more rigorous analysis of the mathematical model.

L13: prefer -> choose a different term here. It's not clear why siderophores are privatized.

L23: are benefitted -> are fitter

L25: meditated -> mediated

L51: why nascent?

L57: This has also been discussed for competition sensing (10.1038/nrmicro2977) and

inversely for quorum sensing (10.1038/ismej.2015.232, 10.1371/journal.pcbi.1004848)

L82: rather than

L136: by using a flow?

L214: tune down -> maybe I missed this, but is the privatization active?

Reviewer #3 (Remarks to the Author):

This is a nicely written and well presented paper on the regulation and evolution of siderophore production in *P. aeruginosa*. This is a very active field, and the paper makes a novel contribution by showing that siderophores are essentially multi-functional, offering individual protection against ROS in addition to their better known collective role as iron chelators. The experimental approaches are innovative, using cutting-edge imaging, and it is good to see the tight integration of mathematical modelling. While there is a lot to like here, I have some concerns about interpretation and lack consideration of relevant previous work:

1. The paper is couched in terms of cooperation but this seems unnecessary to me. The effect they report - retention of siderophores in the cell under ROS stress - provides an individual benefit because siderophores are multifunctional. The argument therefore is essentially that when siderophores are under individual selection (i.e. ROS selection) there is no longer a social dilemma, which is a no-brainer. I think a more straightforward interpretation of these findings is that individual selection can overwhelm social selection on a trait.

2. The terms "environmental stress" and "stress" are too general given that they are specifically referring to ROS stress

3. Previous work has shown that siderophores can help survive ROS stress (e.g. <https://www.ncbi.nlm.nih.gov/pubmed/24586035>) and should therefore be cited and discussed

4. Previous work has shown the opposite effect for a different aminoglycoside gentamicin (<http://www.pnas.org/content/114/3/546.full>) and the contrasting findings should be discussed in the paper

Reviewer #1 (Remarks to the Author):

Review of “Conditional privatization stabilizes bacterial cooperation”

This is a very interesting study demonstrating that cells under environmental stress retain the siderophore, pyoverdinin, at the cell surface, converting pyoverdinin from a normally cooperative factor to a ‘privatized’ factor under these conditions. The results are relevant and interesting, because they provide new insight into cooperation and how cooperative traits are maintained. The results also provide potential explanations of findings from this and other microbial cooperation systems, which will no doubt provide the basis of much future work.

The studies are convincing, presented in a logical order, and for the most part, clear to understand. Most potential issues were addressed. There were only a few minor suggestions:

1. Fig. 1, and the description of Fig. 1 on lines 61-76, was a bit confusing. The line “Fluorescent PVDIs were also excited when illuminated by the 405 nm laser source” (lines 67-68) seems to imply 405 nm illumination AND another approach were used, but the other approach is not described (or perhaps something obvious is being overlooked?). The line “The accumulation of PVDI in the bacterial periplasm” (lines 73-74) should be clarified by adding “under illumination conditions $>3.00 \text{ mW cm}^{-2}$ ”. Also if possible, results with a *pvdA* mutant should be added to Fig. 1 to demonstrate that the method described specifically detects PVDI and not something else. Although such results are presented in Fig. 2, the manuscript would read more clearly if they were also provided in Fig. 1. Also, at what time point were the images and data in Fig. 1c, d and e collected? Presumably, cells from all the compared conditions were taken at the same time but this is not indicated.

Reply 1.1: We thank the reviewer for the helpful suggestions. We revised manuscript to clarify this point (**Page 4: line 68-74**). We added a representative image (**Fig. 1b**) in the revised Figure 1 to show that $\Delta pvdA$ mutant does not accumulate fluorescent substance in their periplasm under the light stimulation ($6.00 \text{ mW} \cdot \text{cm}^{-2}$). We revised the figure caption (**Page 23: line 509-516**) to indicate the time points for the imaging and data collecting of **Figure 1d, e and f**.

2. Fig. 2, as in Fig. 1, the time of illumination needs to be indicated.

Reply 1.2: We thank the reviewer for the helpful suggestion. We revised figure caption (**Page 26: line 531-532**) to indicate the time of illumination.

3. Fig. 3 legend, line 457, “bacteria can tune down the efflux of PVDI” should be clarified with “in correlation with increasing stress.” Also, it is not known that efflux is specifically the mechanism of PVDI release from cells.

Reply 1.3: We thank the reviewer for the thoughtful comment. To directly address this point, we constructed an additional mutant $\Delta pvdRT - ompQ$ deficient in PvdR, PvdT and OmpQ expression. *P. aeruginosa* uses an efflux pump composed of PvdR, PvdT, and OmpQ to secrete PVDI to the environment in a specific manner (**Imperi F, Tiburzi F, & Visca P (2009) PNAS 106(48): 20440-20445**). We found that 1) the initial level of intracellular PVDI of the $\Delta pvdRT - ompQ$ mutant is markedly higher than that of wild-type cells (**Fig. R1 or Fig. S4**), which is consistent with the findings in the literature; and 2) light stimulation ($3.00 \text{ mW} \cdot \text{cm}^{-2}$) did not result in further accumulation of PVDI in $\Delta pvdRT - ompQ$ cells, which is sharply in contrast to the finding that $3.00 \text{ mW} \cdot \text{cm}^{-2}$ illumination results in the accumulation of PVDI within 20 minutes in wild-type cells (**Fig. R1 or Fig. S4**). These findings strongly evidence that *P. aeruginosa* can tune down the efflux pump (PvdRT – OmpQ) in the presence of environmental stresses. To incorporate these results in our manuscript, we have revised the section of **Results (Page 8: line 162-174)** and **Discussion (Page 12: line 256-258)** in the main text.

Figure R1 | Light stimulation does not result in a further accumulation of PVDI in $\Delta pvdRT - ompQ$ strain, where $3.00 \text{ mW} \cdot \text{cm}^{-2}$ violet light (405 nm) was applied to illuminate cells.

4. Fig. 4D – the symbols are missing from the figure legend or not indicated. It is not clear what the numerical numbering in the illustration represents.

Reply 1.4: We thank the reviewer for the useful comment. We revised figure caption (**Page 28: line 554-557**) to indicate these experimental conditions.

5. It seems, based on Fig. 4 and the description of the model, there is a mistake in the figure legend. The range of γ/μ is from 0.1 (selfishness) to 50 (loyal cooperation), not 0.1 (loyal cooperation) to 50 (selfishness) as stated.

Reply 1.5: We thank the reviewer for the carefully reading. We corrected the figure caption (**Page 30: line 565-566**) in the revised manuscript.

6. Fig. 5 – the colors in part B should be changed so they match part A.

Reply 1.6: We thank the reviewer for the helpful suggestion. We recolored the Figure 5b in the revised manuscript.

7. Fig. 5 and lines 192-203. This is very interesting. Why is it that environmental stress seems to result in a greater proportion of cheaters in the models, is this simply due to fitness effects?

Reply 1.7: We thank the reviewer for the excellent question. On the one hand, the secretion of PVDI by the loyal cooperator strain (constitutively secretes of PVDI despite the environmental stress) increased with γ/μ in our model, which gradually in turn repressed the protective activity of intracellular PVDI. On the other hand, our results indicated that public PVDI can also help the cheater strain ($\Delta pvdA$) to survive in the presence of environmental stress (**Fig.2 and Fig. S2**). These findings emphasized that public siderophores can be always exploited by cheaters despite their functions; that is, cheaters can use these siderophores to acquire iron in iron-limited conditions or use public siderophores to protect themselves in the presence of environmental stress. Consequently, the fitness effect as well as the protective effect

of intracellular PVDI enables the cheater strain to outcompete the loyal cooperator strain by constitutively secreting PVDI ($\gamma/\mu = \text{const.} \gg 1$) even in the presence of environmental stress. We have added these findings in the section of **Results** in the revised manuscript (**Page 10: line 193-203**).

Reviewer #2 (Remarks to the Author):

Review “Conditional privatization stabilizes bacterial cooperation”.

The paper “Conditional privatization stabilizes bacterial cooperation” investigates the public goods dilemma arising from siderophore secretion in colonies of *Pseudomonas aeruginosa*. It describes a mechanism by which through reactive oxygen species stress (ROS), the iron-scavenging siderophore pyoverdine (PVDI) becomes “private”, and therefore unexploitable by non-secreting cheaters. The authors conclude that this mechanism that they call “conditional privatization” may be a way to stabilize otherwise fragile cooperative traits.

The paper is written in quite a clear way, and the figures are nice and communicate the results well. First, the authors show that laser excitation and low doses of the antibiotic tobramycin both induce cellular stress through ROS, and that periplasmic accumulation of PVDI can protect cells. The authors then conduct competition assays and employ a mathematical model to show that stress-induced cessation of PVDI secretion---conditional privatization---can allow PVDI producers to outcompete cheaters where constitutive secretors (here called ‘loyal cooperators’) would not.

I think the paper shows a potentially neat mechanism by which PVDI secretion may be inherently protected from exploitation. But to be published, some critical additional data is required, and some of the evolutionary analyses need to be revised.

The authors refer to the observation that stress tends to favor competitive phenotypes, and contrast this with their findings that stress here appears to stabilize the cooperative trait of PVDI secretion through privatization. This is no contradiction as privatizing a cooperative public goods trait as a result of stress appears competitive to me.

Reply 2.1: We thank the reviewer for the carefully reading. We agreed that there is no contradiction between that stress tends to favor competitive phenotypes and stress tends to privatize cooperative public goods. We revised the manuscript to correct this point **(Page 3: line 41-44)**.

To investigate the effect of stress on PVDI secretion and compare fitness between secretors and cheaters, the authors frequently switch between ROS-stress induced by laser and by antibiotic treatment. For example, figure 3 shows that supernatant of bacterial cultures containing increasing concentrations of tobramycin had decreasing amounts of PVDI, and that increased laser intensities led to higher PVDI retention. But while the authors do show that illumination conditions did not affect the growth of cells, such an analysis is missing for the antibiotic treatment. This should be added to clarify

if supernatant concentrations ought to be corrected for cell density, and for the following competition assays.

Reply 2.2: We thank the reviewer for the critical comments and thoughtful suggestions. To directly address this point, we performed additional experiments. First, we examined the growth of cells in the presence of tobramycin at different concentrations, in which the resultant concentration ranged from 0.0 to 2.0 $\mu\text{g} \cdot \text{mL}^{-1}$. We found that the presence of a relatively low dose tobramycin ($\leq 1.0 \mu\text{g} \cdot \text{mL}^{-1}$) did not affect the growth of cells (**Fig. R2 or Fig. S1b**). Then we repeated the experiment by using more rigorous experimental produces; the results showed that PVDI in the supernatant was negatively correlated to the concentration of tobramycin. We added low concentrations of tobramycin to the medium to ensure that tobramycin does not affect the growth of cells, carefully controlled the final bacterial density to ensure that they nearly stayed at an identical level, and obtained a fluorescent photo of supernatants to show the amounts of PVDI. Our new results reconfirmed that PVDI secreted in the supernatant was negatively correlated to the concentration of tobramycin (**Fig. R3 or Fig. 3a**). To clarify this point, we revised the section of **Results (Page 7: line 128-132)**, **Method (Page 15: line 303-304)** and **Figure 3a (Page 27: line 536-538)** in the revised manuscript.

Figure R2 | Growth rate of wide-type (PAO1) in the presence of antimicrobial stresses, where antimicrobial stress is generated by using the treatment of tobramycin.

Figure R3 | Representative regular + fluorescent photos show that the amount of PVDI in the supernatant of bacterial cultures negatively relates to the concentration of tobramycin, where all the bacterial cultures were harvested at an identical bacterial density ($\text{O.D.}_{600} = 0.3$).

The competition assays and mathematical analyses are asking the question: does a PVDI producer (constitutive or conditionally private) end up with a higher frequency than a cheater under different iron limitations (then PVDI becomes an exploitable ‘common good’), and different ROS stress levels. ROS mediated privatization of PVDI occurs at some threshold level through an unidentified mechanism (is there evidence that this is ‘active’ and that transporters are downregulated, or is it simply chemical such that the chelating reaction in the periplasm prevents passage through the outer membrane?).

Reply 2.3: We thank the reviewer for the thoughtful comment. To directly address this point, we constructed an additional mutant $\Delta pvdRT - ompQ$ deficient in PvdR, PvdT and OmpQ expression. *P. aeruginosa* uses an efflux pump composed of PvdR, PvdT, and OmpQ to secrete PVDI to the environment in a specific manner (**Imperi F, Tiburzi F, & Visca P (2009) PNAS 106(48): 20440-20445**). We found that 1) the initial level of intracellular PVDI of the $\Delta pvdRT - ompQ$ mutant is markedly higher than that of wild-type cells (**Fig. R1 or Fig. S4**), which is consistent with the findings in the literature; and 2) light stimulation ($3.00 \text{ mW} \cdot \text{cm}^{-2}$) did not result in further accumulation of PVDI in $\Delta pvdRT - ompQ$ cells, which is sharply in contrast to the finding that $3.00 \text{ mW} \cdot \text{cm}^{-2}$ illumination results in the accumulation of PVDI within 20 minutes in wild-type cells (**Fig. R1 or Fig. S4**). These findings strongly evidence that *P. aeruginosa* can tune down the efflux pump (PvdRT – OmpQ) in the presence of environmental stresses. To incorporate these results in our manuscript, we have revised the section of **Results (Page 8: line 162-174)** and

Discussion (Page 12: line 256-258) in the main text.

Figure R1 | Light stimulation does not result in a further accumulation of PVDI in $\Delta pvdRT - ompQ$ strain, where $3.00 \text{ mW} \cdot \text{cm}^{-2}$ violet light (405 nm) was applied to illuminate cells.

Since privatization of PVDI here appears to have no associated cost, somewhat unsurprisingly then ROS induced privatization leads to an advantage for cooperators. This could be a two-fold benefit: the protection against ROS conferred by PVDI could mean that cheaters die more, or it could be that exploitation is avoided such that cheaters cannot grow under iron limitation. Which one? it is cannot be seen from the data as only proportions of cooperators (both in model and experiment) are shown.

Reply 2.4: We thank the reviewer for the thoughtful comment. To directly address this point, we determined the growth rate of $\Delta pvdRT - ompQ$ strain in different growth conditions, including iron-limited conditions ($\text{FeCl}_3 = 1.0 \times 10^{-5} \mu\text{M}$ or $\text{FeCl}_3 = 0.1 \mu\text{M}$) and iron-limited condition in the presence of exogenous PVDI ($\text{FeCl}_3 = 0.1 \mu\text{M} + \text{ePVDI} = 5.0 \mu\text{M}$). Note that the $\Delta pvdRT - ompQ$ mutant can be regarded as the constitutive privatization strain, in which privatization is independent of the environmental stresses. We found that 1) in the iron-limited condition of $\text{FeCl}_3 = 1.0 \times 10^{-5} \mu\text{M}$ or $\text{FeCl}_3 = 0.1 \mu\text{M}$, the growth rate of ($\Delta pvdRT - ompQ$) mutant ($\mu_3 = 0.0032 \text{ min}^{-1}$ at $\text{FeCl}_3 = 1.0 \times 10^{-5} \mu\text{M}$, **Fig. R4 or Fig. S8** or $\mu_3 = 0.0035 \text{ min}^{-1}$ at $\text{FeCl}_3 = 0.1 \mu\text{M}$, **Fig. R5 or Fig. S9**) was lower than that of the wild-type strain ($\mu_1 = 0.0039 \text{ min}^{-1}$ at $\text{FeCl}_3 = 1.0 \times 10^{-5} \mu\text{M}$, **Fig. R4 or Fig. S8** or $\mu_1 = 0.0099 \text{ min}^{-1}$ at $\text{FeCl}_3 = 0.1 \mu\text{M}$, **Fig. R5 or Fig. S9**); and 2) the

exogenous addition of PVDI (ePVDI = 5.0 μM) markedly increased the growth rate of the ($\Delta pvdRT - ompQ$) strain ($\mu_3 = 0.0100 \text{ min}^{-1}$ at $\text{FeCl}_3 = 1.0 \times 10^{-5} \mu M + \text{ePVDI} = 5.0 \mu M$, **Fig. R5 or Fig. S9**) in iron-limited conditions. This finding indicated that the secretion of private PVDI is beneficial to bacterial growth in iron-limited conditions; and that the PVDI constitutive privatization strain can exploit public PVDI to gain benefits in iron-limited conditions, similar to the non-producer strain. Consequently, the growth of the cooperator strain would become slower when ROS stress triggers the privatization of PVDI if adequate ePVDI is not present in the environment, which is similar to the cheater strain. It should be emphasized that both the cooperator and cheater strains in our competition assay could access extracellular PVDI (public PVDI) even if ROS stress triggered the privatization of PVDI in the cooperator strain. Therefore, these results strongly suggested that PVDI privatization cells outcompete non-producer strains in the presence of ROS stress due to the protective activity of intracellular PVDI.

To further demonstrate that, we directly calculated the kinetics of the extracellular PVDI concentration ($[PVDI]^o(t)$), the growth rate ($\mu_1(t)$ or $\mu_2(t)$), death rate ($p_1(t)$ or $p_2(t)$), and intracellular PVDI concentration ($[PVDI]_1^i(t)$ or $[PVDI]_2^i(t)$) of the cooperator or cheater strain during the competition by using our theoretical model. Our calculations indicated that 1) the cooperator strain outcompetes the cheater strain in the presence of environmental stress (i.e., $\text{FeCl}_3 = 5.0 \times 10^{-3} \mu M$ and Tobramycin = 2.0 $\mu g \cdot mL^{-1}$, **Fig. R6 or Fig. S12**); 2) the domination of the cooperator strain led to the accumulation of extracellular PVDI, resulting in a gradual increase of $[PVDI]^o$ over time (**Fig. R6 or Fig. S12**); 3) an increase in $[PVDI]^o$ led to an increase in the growth rate of both the cooperator and cheater strains (**Fig. R6 or Fig. S12**, note that $\mu_2(t) > \mu_1(t)$ despite their increase); 4) the intracellular PVDI concentration was high in the cooperator strain (**Fig. R6 or Fig. S12**) because ROS stress triggers the privatization of PVDI, which further results in the reduction of the death rate of the cooperator strain; and 5) the intracellular PVDI concentration in the cheater strain remains at a low level in the initial stage and then gradually increases

with increase in $[PVDI]^o$ (note that the cheater strain could also tune down the efflux of PVDI in the presence of environmental stress), which further results in an increase in the death rate of the cheater strain, although it gradually decreases in the later stage.

The additional experimental data and evolutionary analysis together evidence that PVDI privatization cells outcompete nonproducers in the presence of ROS stress due to the protective activity of intracellular PVDI. We clarified this point in the section of **Evolutionary analysis** in the revised **Supplemental materials (Page 3: line 26-40 and Page 5: 69-93)**.

Figure R4 | Growth curves of (a) wild-type (PAO1) and (b) $\Delta pvdRT - ompQ$ strain in the presence of $1.0 \times 10^{-5} \mu M$ $FeCl_3$ at the SSM medium, where the actual growth rate (μ) is determined using a linear fitting (dash line) in the semi-logarithmic plotting.

Figure R5 | Growth curves of (a) wild-type (PAO1) and (b) $\Delta pvdRT - ompQ$ strain in the presence of $0.1 \mu M$ $FeCl_3$ at the SSM medium. Growth curve of (c) $\Delta pvdRT - ompQ$ strain in the presence of $0.1 \mu M$ $FeCl_3$ and $5.0 \mu M$ ePVDI in SSM medium. Actual growth rate (μ) is determined using a linear fitting (dash line) in the semi-logarithmic plotting.

Figure R6 | Kinetics of fraction of wild-type strain (a), extracellular PVDI concentration ($[PVDI]^o(t)$) (b), the growth rate ($\mu_1(t)$ or $\mu_2(t)$) (c), the death rate ($p_1(t)$ or $p_2(t)$) (d), and the intracellular PVDI concentration ($[PVDI]_1^i(t)$ or $[PVDI]_2^i(t)$) (e) of wild-type or $\Delta pvdA$ strain during the competition at the condition of ($FeCl_3 = 5 \times 10^{-3} \mu M$ and Tobramycin = $2.0 \mu g \cdot mL^{-1}$).

PVDI is costly to secrete but enables growth under iron limitation to all cells in the community; it can thus be exploited. But when made private, it is then not available to cheaters any longer and in addition protects cooperators from cell damage and death, improving cooperators relative growth rate over cheaters even when iron is abundant. This begs the question: where is the cost to privatization of the public good? Again, one would need information on total cell numbers / yields from the experiments and the model to investigate the potential cost to not secreting PVDI.

The competition assays and model results are therefore insufficient for the rather sweeping evolutionary claims made towards the end. Cooperation via public good secretion can be evolutionary stable if, for example, colonies of cooperators grow to much higher densities than colonies invaded by cheats. The former would then yield

large numbers of cooperators to the ‘global’ count of cells, while the latter would add only small numbers of cheats. It is therefore important to have an idea of ‘yield’ in competition assays and mathematical models, and it is critical to not only investigate local competition for an evolutionary analysis. Especially the mathematical model clearly lends itself for more thorough evolutionary analysis. For example, one could use an adaptive dynamics framework (e.g. doi:10.1098/rspb.2009.1480, doi:10.1007/BF02409751) to study stress-induced privatization of PVDI, but again, without cost this does not seem like much of a dilemma.

Reply 2.4: We really appreciate the reviewer for this critical comment. To evaluate the cost of privatization of public PVDI, we carefully examined the growth rate of $\Delta pvdRT - ompQ$ strain in different growth conditions (see **Reply 2.4**). The results of the constitutive privatization strain indicated that 1) the production of PVDI is metabolically expensive, irrespective of the produced PVDI being public or private; 2) the cost of PVDI secretion is insufficient to affect the bacterial fitness; and 3) the constitutive privatization strain can also exploit public PVDI to gain benefit. Accordingly, we predicted the evolutionary outputs of the direct competition in different conditions, including the competition between the wild-type (conditional privatization) and $\Delta pvdRT - ompQ$ (constitutive privatization) strains, between $\Delta pvdRT - ompQ$ and $\Delta pvdA$ strains (cheater strain), and among wild-type, $\Delta pvdRT - ompQ$, and $\Delta pvdA$ strains by using our theoretical model (Details are given in **Supplemental materials (Page 3: line 41-67)**). To improve our evolutionary analysis, we evaluated the total number ($N(t)$) of each bacteria strain used in our competition assay, where t is the evolutionary time. Note that $N(t) \equiv \phi(t)V/v$ (see **Method**), where $\phi(t)$ is the volume fraction of bacteria and V/v is a constant. Eq. (11) and (12) (see **Method**) indicated that $\phi(t)$ related to the global dilution rate (δ) set in the experiment. To ensure that $N_r(t)$ (or $\phi_r(t)$) is independent of δ , we normalized $N(t)$ by $N(t)/N_r(t)$ to calculate a relative yield using Eq. (R1)

$$\text{relative yield} = \log_{10}[N(t)/N_r(t)] \quad (\text{R1})$$

where $N_r(t)$ represents the total bacterial number of a reference strain (wild-type strain used for reference) grown under conditions identical to those used in the competition assay (i.e., identical iron abundance, ROS stress and initial bacterial

number). The relative yield can be used to assess whether the tested strain can invade the wild-type strain (the reference strain); namely, the positive or negative relative yield ($\log_{10}[N(t)/N_r(t)] > 0$ or $\log_{10}[N(t)/N_r(t)] < 0$) represents that the invasion of tested strain to the wild-type strain is favorable or unfavorable respectively. Our calculation indicated that 1) $\Delta pvdRT - ompQ$ coexists with the wild-type strain in the presence of ROS stress (**Fig. R7a or Fig. S10a**) and 2) the relative yield is negative in the coexisting region (**Fig. R7b or Fig. S10b**). These results are in contrast with those of the competition of $\Delta pvdA$ and wild-type strain, in which the relative yield is positive in the region dominated by $\Delta pvdA$ cells (**Fig. R7c and d or Fig. S10c and d**). These results indicated that the invasion of $\Delta pvdRT - ompQ$ to wild-type strain is evolutionary unfavorable in the presence of ROS stress. To further demonstrate this, we predicted the evolutionary outputs of the direct competition among the wild-type (*conditional privatization*), $\Delta pvdRT - ompQ$ (constitutive privatization) and $\Delta pvdA$ (cheater strain) strains. **Figure S11a and b (Fig. R8)** shows that the wild-type strain is evolutionary favorable in the presence of ROS stress; that is, the wild-type strain cannot be invaded by neither $\Delta pvdRT - ompQ$ nor $\Delta pvdA$ strain in the presence of ROS stress.

We clarified this point in the section of **Evolutionary analysis** in the revised **Supplemental materials (Page 3: line 41-67)**.

We revised the section of **Results** in the main text (**Page 10: line 214-216**) and cited this reference mentioned by the reviewer.

Figure R7 | Fractions (a) or the relative yields (b) of $\Delta pvdRT - ompQ$ as a function of the abundance of ferric and the normalized ROS stress in the directly competition of $\Delta pvdRT - ompQ$ and wild-type strain. Fractions (c) or the relative yields (d) of $\Delta pvdA$ as a function of the abundance of ferric and the normalized ROS stress in the directly competition of $\Delta pvdA$ and wild-type strain. Evolution time is 30 days.

Figure R8 | Fractions or the relative yields of $\Delta pvdA$ (a, b), $\Delta pvdRT - ompQ$ (c, d) or wild-type strain (e) as a function of the abundance of ferric and the normalized ROS stress in the directly competition of $\Delta pvdRT - ompQ$, $\Delta pvdA$ and wild-type strain. Evolution time is 30 days.

Finally, I would prefer removing the discussion section on evolution in human societies, as this appears to be too much of a stretch for what is not really an evolutionary analysis.

Reply 2.5: We appreciate the reviewer for this kind suggestion. We agree that the discussion on evolution human societies is much of a stretch. This kind of discussion is motivated to inspire the readers to think the potential principles by which underlying the evolution of cooperation. For this reason, we would prefer reserving this discussion. However, we would accept the reviewer suggestion and remove this discussion in the next round revision if the reviewer insists her/his opinion.

Also there are some minor issues I noticed that I am attaching as a list at the end (non-exhaustive).

I sum, I think this paper has much potential due to the thorough investigation of the PVDI phenotype as a function of ROS. To be published in Nature Communication and to justify the evolutionary claims, however, requires more work, if at least via more rigorous analysis of the mathematical model.

L13: prefer -> choose a different term here. It's not clear why siderophores are privatized.

L23: are benefitted -> are fitter

L25: meditated -> mediated

L51: why nascent?

L57: This has also been discussed for competition sensing (10.1038/nrmicro2977) and inversely for quorum sensing (10.1038/ismej.2015.232, 10.1371/journal.pcbi.1004848)

L82: rather than

L136: by using a flow?

Reply 2.6: We thank the reviewer for the carefully reading. We correct these points in the revised manuscript (**Page 2: line 23, Page 2: line 25, Page 3: line 34-35, Page 3: line 41-42, Page 3: line 51, Page 4: line 83, Page 8: line 147-148**). We also revised the section of **Introduction** a bit in the main text (**Page 2: line 13**) and cited these references mentioned by the reviewer.

L214: tune down -> maybe I missed this, but is the privatization active?

Reply: See the reply 2.3.

Reviewer #3 (Remarks to the Author):

This is a nicely written and well presented paper on the regulation and evolution of siderophore production in *P. aeruginosa*. This is a very active field, and the paper makes a novel contribution by showing that siderophores are essentially multi-functional, offering individual protection against ROS in addition to their better known collective role as iron chelators. The experimental approaches are innovative, using cutting-edge imaging, and it is good to see the tight integration of mathematical modelling. While there is a lot to like here, I have some concerns about interpretation and lack consideration of relevant previous work:

1. The paper is couched in terms of cooperation but this seems unnecessary to me. The effect they report - retention of siderophores in the cell under ROS stress - provides an individual benefit because siderophores are multifunctional. The argument therefore is essentially that when siderophores are under individual selection (i.e. ROS selection) there is no longer a social dilemma, which is a no-brainer. I think a more straightforward interpretation of these findings is that individual selection can overwhelm social selection on a trait.

Reply 3.1: We thank the reviewer for the critical comments. It should be emphasized that the protective activity of intracellular siderophores plays an essential role in— but is insufficient in—maintaining the bacterial cooperation. Our results indicated that the exogenous addition of PVDI also enables the cheater strain ($\Delta pvdA$) to survive in the presence of environmental stress (**Fig.2 and Fig. S2**). These findings indicate that public siderophores can be always exploited by cheaters despite their functions; that is, cheaters can use these siderophores to acquire iron in iron-limited conditions or use siderophores to protect themselves in the presence of ROS stress. Furthermore, theoretical calculations predicted that the cheater strain could outcompete loyal cooperators ($\gamma/u \gg 1$, **Fig. 5a**) even in the presence of strong ROS stress. This result indicates that *conditional privatization* is an essential mechanism to maintain cooperation rather than ROS selection. We revised the section of **Results** in the main text to clarify this point (**Page 10: line 193-203**).

2. The terms "environmental stress" and "stress" are too general given that they are specifically referring to ROS stress

Reply 3.2: We thank the reviewer for the helpful suggestion. We replaced the

‘Environmental stress’ in the **Fig. 4b, 5a and 5b** by ‘ROS stress’.

3. Previous work has shown that siderophores can help survive ROS stress (e.g. <https://www.ncbi.nlm.nih.gov/pubmed/24586035>) and should therefore be cited and discussed

Reply 3.3: We thank the reviewer for the helpful suggestion. We revised the section of **Results** in the main text (**Page 5: line 92-93**) and cited this reference mentioned by the reviewer.

4. Previous work has shown the opposite effect for a different aminoglycoside gentamicin (<http://www.pnas.org/content/114/3/546.full>) and the contrasting findings should be discussed in the paper

Reply 3.4: We appreciate the reviewer for this critical comment. We agree that there is an obvious contradiction between our findings and those of Vasse’s findings (**Vasse M, et al. (2017) PNAS 114(3):546-551**). To explain this contradiction, firstly, we carefully compared the details of their and our experiments (**see their Supplementary Materials**) and found that the major difference is that they used the CAA mediums to perform all experiments, whereas we used the SSM medium. Note that the SSM medium (we used) is a well-defined minimal medium, which enabled us to rigorously control the ferric concentration in the medium to generate iron-limited conditions. However, the CAA medium (they used) contains casamino acid, which is a mixture of amino acids and some very small peptides obtained from the acid hydrolysis of casein; thus, the CAA medium is an iron-rich medium. Consequently, Vasse et al. had to add apotransferrin (20 mM) in the CAA medium to generate iron-limited conditions. It should be emphasized that iron abundance in the medium is a key factor that determines whether PVDI can protect cells and enable them to survive in the presence of antimicrobial stress, presumably because intracellular PVDI can repress the Fenton reaction through the chelation of free ferric and ferrous. To confirm this, we investigated whether PVDI can protect cells against ROS stress in iron-rich conditions. Our results indicate that PVDI cannot protect cells against ROS

stress any more in the iron-rich medium, including the SSM medium + 100 μM FeCl_3 (**Fig. R9 or Fig. S3**) or pure CAA medium (**Fig. R9 or Fig. S3**). Furthermore, we examined whether PVDI can protect cells against ROS stress generated by gentamicin ($2.0 \mu\text{g} \cdot \text{mL}^{-1}$) in iron-limited conditions (SSM + $10^{-5} \mu\text{M}$ FeCl_3) (**Fig. R10 or Fig. S2**). We found that PVDI protects cells against ROS stress generated by gentamicin in the CAA medium + 20 mM apotransferrin (**Fig. R10 or Fig. S2**); this finding is consistent with our previous result (**Fig. S2 a and b**) that PVDI can protect cells against ROS stress generated by tobramycin in iron-limited conditions. This consistency suggests that the use of intracellular PVDI by *P. aeruginosa* to protect themselves against ROS stress is a general mechanism.

We noticed that the bacterial growth was very slow in the CAA medium containing 20 mM apotransferrin in the first 12 hours, indicating that the availability of iron was limited. However, the growth rate of bacteria increased in this medium after 20 hours of culturing and the bacterial culture eventually reached a very high density (OD = 1.025, 44 hours); this finding is in agreement with that of Vasse et al.. This phenomenon suggests that the availability of iron was not limited in the later culturing stage, which is presumably because *P. aeruginosa* can gradually secrete proteases that may degrade the apotransferrin (**Shigematsu T, et al. (2001) Microbiol. Immunol. 45(8):579-590.**), thus releasing chelating irons in the later stage. To validate this hypothesis, we examined whether PVDI can protect cells against ROS stress generated by gentamicin ($2.0 \mu\text{g} \cdot \text{mL}^{-1}$) in the CAA medium + 20 mM apotransferrin in the later stage (24 hours). Our results indicate that PVDI **cannot** protect cells against ROS stress any more in the later culturing stage (24 hours) (**Fig. R10 or Fig. S3**), suggesting that the availability of iron was not limited in the later culturing stage. Most importantly, the competition assay employed by Vasse et al. was performed for 48 hours (**see their Supplementary Materials**), thus, the iron-limited conditions might have not retained in their assay. This could further lead to intracellular PVDI not being to protect cells in the presence of antimicrobial stress. To evaluate how protection activity arose from intracellular PVDI affects final

evolutionary outputs, we performed additional theoretical calculations. Our results confirmed that cheater always outcompetes the cooperator strain despite ROS stress if intracellular PVDI cannot protect cells (**Fig. R11 or Fig. S13**), which generally agrees with Vasse's findings. We revised the section of **Results (Page 6: line 102-107)**, **Method (Page 15: line 297-315)** and **Supplementary Materials (Page 5: line 86-93)** to clarify this point.

Figure R9 | PVDI protect cells to survive in the presence of antimicrobial stresses at iron-limited conditions. Treatment of $1.0 \mu\text{g} \cdot \text{mL}^{-1}$ tobramycin (7 h) lead more fraction $\Delta pvdA$ cells generated ROS than that of wild-type cells (a), more fraction $\Delta pvdA$ cells to damage than that of wild-type cells (b) in iron-limited condition (SSM + $\text{FeCl}_3 = 1.0 \times 10^{-5} \mu\text{M}$). Treatment of $2.0 \mu\text{g} \cdot \text{mL}^{-1}$ gentamicin (7 h) lead more fraction more fraction $\Delta pvdA$ cells generated ROS than that of wild-type cells, more fraction $\Delta pvdA$ cells to damage than that of wild-type cells in iron-limited condition, including (c, d) SSM + $\text{FeCl}_3 = 1.0 \times 10^{-5} \mu\text{M}$ or (e, f) CAA + 20 mM apotransferrin (fresh).

Figure R10 | PVDI cannot protect cells to survive in the presence of antimicrobial stresses at iron-rich conditions. Cell fraction (a) generated ROS or (b) damaged arising from $\Delta pvdA$ strain is similar to that arising from wild-type strain after treatment of $1.0 \mu g \cdot mL^{-1}$ tobramycin (7 h) at iron-rich condition (SSM + $FeCl_3 = 100 \mu M$). Cell fraction generated ROS or damaged arising from $\Delta pvdA$ strain is similar to that arising from wild-type strain after treatment of $2.0 \mu g \cdot mL^{-1}$ gentamicin (7 h) at iron-rich conditions, including (c, d) (SSM + $FeCl_3 = 100 \mu M$), (e, f) CAA or (g, h) CAA + 20 mM apotransferrin that had been used to culture bacterial strain for 24 h.

Figure R11 | Fractions of wild-type strain as a function of the abundance of ferric and the normalized ROS stress in the directly competition of $\Delta pvdA$ and wild-type strain, where intracellular PVDI were assumed to not be able to protect cells survive in the presence of ROS stress.

REVIEWERS' COMMENTS:

Reviewer #1 (Remarks to the Author):

The revisions address all concerns raised in the first review and is now acceptable for publication as is. The edited manuscript is much clearer and greatly improved.

Reviewer #2 (Remarks to the Author):

The authors have put in a lot of work to address the reviewers comments. In particular, additional experiments clarified the mechanism of PVDI privatization (preventing exploitation vs. protection from death). Additional modeling made the evolutionary analyses more complete. I support publication.

Reviewer #3 (Remarks to the Author):

The authors have done an outstandingly thorough job of responding to my comments on the previous version. They have fully addressed each point and I am completely satisfied. I recommend that the revised manuscript be accepted.